# Distributed Distillation for On-Device Learning

**Ilai Bistritz, Ariana J. Mann, Nicholas Bambos**
Stanford University
{bistritz,ajmann,bambos}@stanford.edu

## Abstract

On-device learning promises collaborative training of machine learning models across edge devices without the sharing of user data. In state-of-the-art on-device learning algorithms, devices communicate their model weights over a decentralized communication network. Transmitting model weights requires huge communication overhead and means only devices with identical model architectures can be included. To overcome these limitations, we introduce a distributed distillation algorithm where devices communicate and learn from soft-decision (softmax) outputs, which are inherently architecture-agnostic and scale only with the number of classes. The communicated soft-decisions are each model's outputs on a public, unlabeled reference dataset, which serves as a common vocabulary between devices. We prove that the gradients of the distillation regularized loss functions of all devices converge to zero with probability 1. Hence, all devices distill the entire knowledge of all other devices on the reference data, regardless of their local connections. Our analysis assumes smooth loss functions, which can be non-convex. Simulations support our theoretical findings and show that even a naive implementation of our algorithm significantly reduces the communication overhead while achieving an overall comparable accuracy to the state-of-the-art. By requiring little communication overhead and allowing for cross-architecture training, we remove two main obstacles to scaling on-device learning.

## 1   Introduction

Today, software applications and users' personal data are located on the individual's wireless edge (e.g., smartphone or IoT) device, while the machine learning algorithms deployed in these applications relegate their training to central servers. This necessitates the transfer of the users' data to the central server for training, which entails massive communication overhead and privacy issues, limiting the scalability of model training. On the other hand, one device alone does not have enough data to achieve state-of-the-art performance. The goal of on-device learning is to keep both the data and the training on the device by coordinating collaborative training across devices. Achieving this goal will unlock an unprecedented amount of training data that today is generated on user devices.

The first step was accomplished in Federated Learning (FL) [1–3] where each device trains its deep neural network (DNN) with stochastic gradient descent (SGD) on its personal data. The resulting parameters of each DNN are then uploaded to a server where they are averaged and the average is transmitted to every device involved for another round of training. This way, each device's DNN learns from the private data of its peers without ever seeing it. Beyond maintaining privacy, this method has the efficiency of a learning algorithm trained over a huge amount of data without having to ever communicate the often multi-media data itself. However, FL's repeated communication of large parameter vectors is infeasible for power-limited edge devices and its central parameter server is a singular point of failure that can only handle up to a certain number of devices.

To remove the bottleneck of a single server, distributed (i.e device-to-device) SGD based DNN training methods have been studied [4–7]. However, these schemes distribute the computation assuming all

devices can access the entire dataset. The distributed equivalents of FL, with data parallelism, were studied in [8–10]. We broadly refer to these algorithms as Distributed-SGD (D-SGD). The theoretical basis for these D-SGD algorithms is the seminal works on distributed optimization [11–13], which were later generalized to non-convex cost functions in [14, 15]. Hence, by design, all these methods transmit and average model parameters (i.e., DNN weights and biases) such that they can only include in the training process devices that have the same model architecture and the communication overhead is large due to the large models in modern applications [16].

To design a new approach, we want to rethink how DNNs can efficiently learn from each other. Ensemble methods have addressed a similar question by making final predictions from the weighted average of several DNNs' predictions [17]. The work in [18] distills the knowledge of an ensemble of DNNs into one DNN by training a small, student DNN from the soft-decision outputs of an ensemble of large, trained teacher DNNs. Knowledge distillation was introduced in [19] which used one teacher DNN, incorporated labels into the soft-decision based training, and adjusted the temperature in the soft-max function all to positive effects. Soft-decisions are the soft-max of the classification model's logit vector. As a result, learning via soft-decisions can occur between classifiers with different model architectures, which is not possible in learning via weights (i.e., ResNet14 can teach logistic regression, and AlexNet can teach ResNet14) [18]. Additionally, soft-decisions have been shown to be more robust to staleness than weights [20].

In a network of devices there is no teacher, only many students who must learn from each other. Co-distillation methods have successfully, jointly trained small numbers of students over a fully-connected (ie., complete) communication graph [20, 21]. These results empirically demonstrated that co-distillation between two small student DNNs can outperform the teacher-student hierarchy [21] and achieve comparable performance to a DNN ensemble [20]. However, in [21] all DNNs have the same data and communicate every soft-decision. While in [20], they use data parallelism but communicate every DNN's weights so that each local node can evaluate its own data's soft-decisions produced by every other node's DNN and integrate them in training regularization. Both of these methods are concerned with improving the accuracy of a DNN trained in one silo, which does not have the communication constraints or privacy restrictions of the on-device setting.

## 1.1 Distributed Distillation Algorithm

To enable distillation for on-device learning, we introduce Distributed Distillation (D-Distillation), where the students are no longer nodes in the same server but different data silos. Our work is mainly motivated by the setting of mobile and IoT edge devices belonging to different users. In this setting, students have non-overlapping user data that cannot be shared, and every device is connected to only a few nearby devices. These devices are typically battery-limited so it is necessary to reduce the communication overhead and thereby the transmission power consumption.

To maintain a basic level of privacy and avoid massive communication, devices never share any of their private data in D-Distillation. Instead, to generate the soft-decisions that enable distillation, we deploy a public, unlabeled reference dataset. Each device may receive this dataset as a part of the algorithm's software that all participating devices must receive, which leaves open the possibility of a custom-designed and selected dataset. Alternatively, public data points some devices have access to can be propagated. Either way, the reference dataset is simple to construct since unlabeled data points are far easier to collect than labeled data points. Furthermore, a semi-supervised learning regime with a subset of unlabeled data has been beneficial for other deep learning goals such as adversarial robustness [22]. This reference data set serves as a common vocabulary for the devices to share what they learned from their private data without ever sharing the data itself.

D-Distillation is a distributed algorithm that works on a strongly connected, directed communication network where each device communicates with its neighbors. Hence, devices no longer have immediate access to the whole communication network's average soft-decisions as assumed in the classic distillation literature [18–20, 23]. This constitutes the main technical challenge in our analysis.

We prove that our D-Distillation algorithm converges such that the gradients of the distillation loss functions of all devices converge to zero with probability 1. This proves that all the knowledge in the network on the reference data is distilled into every device in the network. Our algorithm communicates only network soft-decisions and can naturally incorporate devices with different DNN architectures, or even other types of classifiers. We demonstrate in simulations with DNNs that a

vanilla implementation of D-Distillation significantly reduces the communication overhead while achieving comparable accuracy to weight sharing methods, which varies by regime. This makes our algorithm appealing for wireless applications where communication bandwidth and transmission power are scarce resources. It leaves open for future work improvements that can be gained in practice by a detailed design of the parameters our algorithm introduces.

## 1.2 Previous Work

FL's introduction in [1] spurred the development of algorithms to mitigate the huge overhead that communicating weights entails. Communication reduction schemes include reducing the frequency of communication relative to training rounds, sketching, sparsification, and quantization [3, 7, 24–31]. While our algorithm also reduces communication, it achieves this by changing the object that is communicated, which in turn changes the type of stationary points our algorithm converges to. Not only can previously developed communication reduction schemes be applied to D-Distillation's network soft-decisions, but D-Distillation introduces the tunable communication parameter of network batch size: the number of network soft-decisions communicated in each training iteration. As a result, it does not directly compete with these communication reduction schemes, but instead can be combined with them (as demonstrated in Fig. 4). It is only the combination of novel techniques such as ours with compression methods that can reduce communication by a factor of millions, which is crucial to enable on-device learning for bandwidth and power-limited wireless devices.

While our algorithm is a distributed extension of knowledge distillation, it also can be thought of as an extension of co-regularization [32, 33]. Distillation requires sharing the private datapoints to communicate soft-decisions, which D-Distillation overcomes by employing a reference dataset of publicly available, unlabeled data. This idea has some resemblance to the bootstrap methods of co-training and co-Expectation-Maximization (co-EM) where two (Naive Bayes or SVMs) classifiers train on disjoint sets of labeled data, provide (noisy) labels on unlabeled data, and then train on the expanded labeled dataset [34, 35]. Co-regularization, similar to distillation, learns from classifier provided labels on unlabeled data through a regularization term in the loss function, instead of augmenting the dataset like in standard co-training. Our distillation-based algorithm shares soft-decisions as opposed to labels as in co-regularization. In any case, our approach generalizes aspects of distillation and co-regularization to the large-scale decentralized network setting with limited communication between many devices. As such, our main focus is analyzing the convergence of D-Distillation over the network and studying the resulting performance.

Published in the same proceedings as this paper, [36, 37] integrate distillation techniques to show promising empirical improvements for FL, for the case of one central server. In [36] they enable cross-architecture learning by communicating model parameters to a central server, performing FedAvg within model types, then distillation across model types, and finally communicating the updated model parameters back to each device. This approach has the same communication overhead as FL and is not suitable for a decentralized network. To improve the computational efficiency, [38] integrates distillation as a mediator between FL and Split Learning [37].

## 2 Problem Formulation

Consider the classification task where the goal is to learn a function that maps every input data point to the correct class out of $K$ possible options. Each device $n$ has access to its own private data $\mathcal{D}_n = \{\tilde{x}_i^n\}_{i=1}^{M_n}$ of inputs and their corresponding labels $\boldsymbol{y}(\tilde{x}_i)$. All labels $\boldsymbol{y}(\tilde{x}_i)$ are hard-decision, one-hot vectors over the set of classes. We emphasize that devices do not have access to the private data of other devices, and private data points are never communicated. Instead, devices communicate with respect to a reference dataset of *unlabeled* data points, denoted $\mathcal{D}_r = \{x_j\}_{j=1}^Q$, that all devices have access to. In this sense our distributed training algorithm is semi-supervised; namely, private learning is done on labeled data and distributed learning is done on unlabeled data.

Each device $n$ has a classification model (e.g., DNN) with $p_n$ parameters (e.g., weights), $\boldsymbol{\theta}^n \in \mathbb{R}^{p_n}$. All these classification models produce a logit vector to which a soft-max function is applied. This results in a probability vector over the $K$ classes, which is the output of the classification model. The class with the highest probability is the learned class for the input. We will refer to the probability vector over the classes as the soft-decision $s_n(\boldsymbol{\theta}^n, x) : \mathbb{R}^{p_n} \times (\mathcal{D}_n \cup \mathcal{D}_r) \to \Delta^K$. We assume that for every $x$ and $n$, the soft-decision function $s_n(\boldsymbol{\theta}^n, x)$ with respect to $\boldsymbol{\theta}^n$ is $L_s$-Lipschitz continuous

(as a vector-valued function) and has $L_g$-Lipschitz continuous gradients (as a matrix-valued function). We use the Euclidean norm for vectors and its induced spectral norm for matrices. This amounts to assuming smoothness of the classification model. Note that we do not assume that $s_n$ is convex. With a slight abuse of notation, we use $s$ instead of $s_n$ when the index is clear through $\boldsymbol{\theta}^n$.

Define the loss function $\mathcal{L}(s, \boldsymbol{y}) : \Delta^K \times \Delta^K \to \mathbb{R}_0^+$. We define the local loss function of device $n$ as

$$\mathcal{L}_n\left(\boldsymbol{\theta}^n\right) \triangleq \sum_{\tilde{x} \in \mathcal{D}_n} \mathcal{L}\left(s\left(\boldsymbol{\theta}^n, \tilde{x}\right), \boldsymbol{y}\left(\tilde{x}\right)\right). \tag{1}$$

Since $s\left(\boldsymbol{\theta}^n, \tilde{x}\right)$ and $\boldsymbol{y}\left(\tilde{x}\right)$ are bounded as probability vectors, then $\mathcal{L}_n\left(\boldsymbol{\theta}^n\right)$ is $L_a$-Lipschitz continuous with Lipschitz continuous gradients for some $L_a$ (for details, see Lemma 10). We note that our results hold for any $\mathcal{L}_n\left(\boldsymbol{\theta}^n\right)$ that is Lipschitz continuous and has Lipschitz continuous gradients. However, $\mathcal{L}_n\left(\boldsymbol{\theta}^n\right)$ does not have to be convex and our results hold for non-convex loss functions.

Let $\mathcal{X}$ be the unknown distribution of a labeled data point $(\tilde{x}, \boldsymbol{y}(\tilde{x}))$. The goal of distributed training algorithms is to learn the parameters $\boldsymbol{\theta}^1, ..., \boldsymbol{\theta}^N$ that minimize the total generalization error in the network: $\mathcal{L}^* \triangleq \mathbb{E}_{\tilde{x} \sim \mathcal{X}}\left\{\sum_{n=1}^N \mathcal{L}\left(s\left(\boldsymbol{\theta}^n, \tilde{x}\right), \boldsymbol{y}\left(\tilde{x}\right)\right)\right\}$. Let $\boldsymbol{\theta} = \left[\boldsymbol{\theta}^1 \cdots \boldsymbol{\theta}^N\right] \in \mathbb{R}^{\sum_{n=1}^N p_n}$ be the concatenated vector of all weights. As with any supervised learning, since the distribution $\mathcal{X}$ is unknown, we try to fit the model $\boldsymbol{\theta}$ that minimizes $\mathcal{L}^*$ based on training samples. With on-device learning, the challenge is to minimize $\mathcal{L}^*$ based on the private training samples of all devices, without actually sharing them between devices. To that end, we propose the following training objective:

**Definition 1.** Let $\{\mathcal{L}_n\}_n$ be the local loss functions. Let $\mathcal{D}_r$ be the set of size $Q$ unlabeled reference data. Define the distillation loss function of device $n$ as [20]:

$$\mathcal{L}_{\mathrm{dist},n}\left(\boldsymbol{\theta}\right) = \mathcal{L}_n\left(\boldsymbol{\theta}^n\right) + \rho \sum_{x \in \mathcal{D}_r} \left\| \frac{1}{N} \sum_{m=1}^N \boldsymbol{s}\left(\boldsymbol{\theta}^m, x\right) - \boldsymbol{s}\left(\boldsymbol{\theta}^n, x\right) \right\|^2 \tag{2}$$

where $\rho > 0$ is a regularization parameter. Our objective is to minimize

$$\min_{\boldsymbol{\theta}} \sum_{n=1}^N \mathcal{L}_{\mathrm{dist},n}\left(\boldsymbol{\theta}\right). \tag{3}$$

The second term in (2) is a distillation regularization term that facilitates each device learning from all other devices in the network. We focus on the L2 loss for the regularization term since our analysis requires the loss function to be smooth in both the variables inside the L2 loss.

This objective in (3) is what co-distillation [20] would minimize in our on-device scenario if we had a fully-connected (complete) graph. However, in a decentralized network (e.g., mobile network, IoT) every device can communicate with only several others based on its position in space, available infrastructure, or social network. Therefore, the main challenge that we overcome in this work is to design a distributed algorithm that provably distills the information from all over the network for every strongly connected, directed communication graph, despite the fact that devices are only connected to their neighbors on this graph. We now define our communication graph and communication matrix, whose values are the weights devices assign to their peers to incorporate their received information.

**Definition 2.** Let $G = (\mathcal{V}, \mathcal{E})$ be the directed communication graph, where $\mathcal{V} = \{1, ..., N\}$ and device $n$ can send messages to device $m$ if and only if $(n, m) \in \mathcal{E}$. We assume that $G$ is strongly connected (i.e., connected when $G$ is undirected). Define the communication matrix $W = \{w_{n,m} \geq 0\}$ of $G$ such that $w_{n,m} > 0$ if and only if $(n, m) \in \mathcal{E}$. We assume that $W$ is doubly stochastic, so $\sum_{m=1}^N w_{n,m} = 1$ and $\sum_{m=1}^N w_{m,n} = 1$ for each $n$, and that $w_{n,n} > 0$ for all $n$.

## 3 Distributed Distillation Algorithm

By communicating over the graph, each device can receive its neighbors' soft-decisions on the reference data. However, to minimize (2) a device has to know the soft-decisions of all devices. To overcome this difficulty, our algorithm solves the following problem instead of directly solving (3)

$$\min_{\substack{\boldsymbol{\theta}^1, ..., \boldsymbol{\theta}^N \\ \boldsymbol{z}^1, ..., \boldsymbol{z}^N}} \sum_{n=1}^N \left[ \mathcal{L}_n\left(\boldsymbol{\theta}^n\right) + \rho \sum_{x \in \mathcal{D}_r} \left\| \boldsymbol{z}^n\left(x\right) - \boldsymbol{s}\left(\boldsymbol{\theta}^n, x\right) \right\|^2 \right] \tag{4}$$

$$\text{s.t. } \boldsymbol{z}^n\left(x\right) = \boldsymbol{z}^m\left(x\right), \ \forall\, (m, n) \in \mathcal{E} \text{ and } \forall x \in \mathcal{D}_r.$$

The form in (4) allows for a distributed algorithm since devices only need to communicate with their neighbors to satisfy the constraints. The learning process of each device, which consists of taking SGD steps on its local regularized loss function, is now only coupled to other devices' learning processes through the auxiliary variable $\boldsymbol{z}^n(x)$, which we call the *network soft-decision*. Since the graph $G$ is strongly connected, the constraint in (4) implies that $\boldsymbol{z}^n(x) = \boldsymbol{z}^m(x)$ for every $x \in \mathcal{D}_r$ and every pair of devices $n, m$. We employ a distributed consensus step and show that by solving (4), the devices learn a solution to (3).

---

**Algorithm 1** Distributed Distillation

---

**Initialization:** Let $\boldsymbol{\theta}_0^n$ and $\boldsymbol{z}_0^n$ be the initial variables of device $n$. Let $\{\mathcal{S}_t^r\}_t$ be a common random sequence of reference data, such that $\mathcal{S}_t^r \subseteq \mathcal{D}_r$ and $|\mathcal{S}_t^r| = b$ for all $t$, for some $b \geq 1$. Let $W = \{w_{n,m}\}$ be a doubly stochastic communication matrix with positive diagonal elements. Let $\beta > 0$. Let the step size sequence be $\{\eta_t\}_t$ and $T$ be the number of training epochs.
**For $t = 1, \ldots, T$ each device runs:**

1. **Communication:** For every $x \in \mathcal{S}_t^r$, broadcast $\boldsymbol{z}_t^n(x)$ to all $m$ such that $(n, m) \in \mathcal{E}$ and receive $\boldsymbol{z}_t^m(x)$ from all $m$ such that $(m, n) \in \mathcal{E}$.

2. **Training**: Using a random subset of private data $\mathcal{S}_t^p \subseteq \mathcal{D}_n$ and $\mathcal{S}_t^r$, update the classifier:

$$\boldsymbol{\theta}_{t+1}^n = \boldsymbol{\theta}_t^n - \eta_t \frac{M_n}{|\mathcal{S}_t^p|} \sum_{\tilde{x} \in \mathcal{S}_t^p} \nabla \mathcal{L}\left(s\left(\boldsymbol{\theta}_t^n, \tilde{x}\right), \boldsymbol{y}\left(\tilde{x}\right)\right)$$

$$- 2\eta_t \beta \frac{Q}{N |\mathcal{S}_t^r|} \sum_{x \in \mathcal{S}_t^r} \left[ \left(\nabla s\left(\boldsymbol{\theta}_t^n, x\right)\right)^T \left(s\left(\boldsymbol{\theta}_t^n, x\right) - \boldsymbol{z}_t^n\left(x\right)\right) \right] \quad (5)$$

3. **Consensus on Network Soft-Decisions**: For each $x \in \mathcal{S}_t^r$, update:

$$\boldsymbol{z}_{t+1}^n(x) = \sum_{m=1}^{N} w_{m,n} \boldsymbol{z}_t^m(x) - 2\beta\eta_t \left(\boldsymbol{z}_t^n(x) - s\left(\boldsymbol{\theta}_t^n, x\right)\right). \quad (6)$$

**End**

---

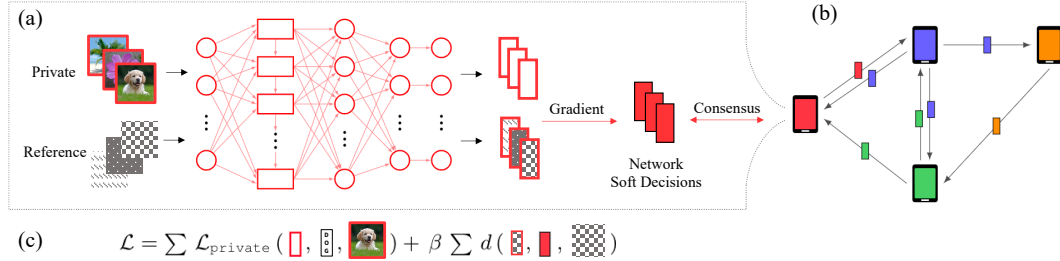

(c) $\mathcal{L} = \sum \mathcal{L}_{\text{private}}(\,\square\,,\,\square\,,\,\square\,) + \beta \sum d(\,\square\,,\,\square\,,\,\square\,)$

Figure 1: In D-Distillation, (a) on each device is its DNN and private data. We provide all devices with the same reference dataset. The device evaluates the DNN on its private data and the reference data to produce its private soft-decisions and its reference soft-decisions respectively. (b) At each iteration, every device communicates a subset of its current network soft-decisions and receives its neighbors' corresponding network soft-decisions. The device then updates its network soft-decisions via a consensus step, combined with a gradient step to better match its own reference soft-decisions. (c) Each device updates its weights with an SGD step on its local regularized loss function.

Our algorithm is summarized in Algorithm 1, and the idea behind it is depicted in Fig. 1. Similarly to DSGD algorithms, the doubly stochastic communication matrix $W$ is agreed upon in advance by the governing application or distributedly between the devices (for details, see [11, 39]).

Algorithm 1 uses the common random sequence of reference data subsets $\{\mathcal{S}_t^r\}_t$ such that $\mathcal{S}_t^r \subseteq \mathcal{D}_r$ for all $t$. We emphasize that our algorithm requires the devices to communicate at each iteration only a small random subset of the reference dataset $\mathcal{D}_r$ (as small as one datapoint). This is analogous to the small random subset of data used for training in batch-SGD. In fact, the size of this subset is a convenient way to control the communication overhead of our algorithm, as can be seen in Section 5.

Devices can obtain $\{\mathcal{S}_t^r\}_t$ in advance from the application that governs the distributed training. Alternatively, $\{\mathcal{S}_t^r\}_t$ can be generated from any common randomness that is available to all devices (e.g., common clock). In case a fully distributed solution is needed, each device can generate a candidate sequence, and add a random identifier from a Gaussian distribution to create a packet. Then devices propagate only the packet with the maximal identifier they possess, and after a bounded time (the diameter of the graph), all devices will agree on a common $\{\mathcal{S}_t^r\}_t$ sequence.

Our main theorem provides convergence guarantees for Algorithm 1. The proof is postponed to Section 10 of the Appendix.

**Theorem 3.** *Assume Algorithm 1 is initialized with $\beta > 0$ and step-size sequence $\{\eta_t\}_t$ such that $\sum_{t=1}^{\infty} \eta_t = \infty$, $\sum_{t=1}^{\infty} \eta_t^2 < \infty$, and $\lim_{t\to\infty} \eta_{t+1}/\eta_t$ exists. Assume that Definition 2 on the communication graph holds. Assume that the soft-decision functions $\{s_n\}$ are Lipschitz continuous and have Lipschitz continuous gradients. Assume that the loss function $\mathcal{L}(s, \boldsymbol{y}) : \Delta^K \times \Delta^K \to \mathbb{R}_0^+$ is twice continuously differentiable. If Algorithm 1 is run, then the gradient of the distillation regularized loss function (2) of each device converges to zero with probability 1, i.e., for every $n$,*

$$\lim_{t\to\infty} \left\| \nabla_{\boldsymbol{\theta}^n} \left[ \mathcal{L}_n\left(\boldsymbol{\theta}_t^n\right) + \frac{\beta}{N-1} \sum_{x \in \mathcal{D}_r} \left\| \frac{1}{N} \sum_{m=1}^{N} \boldsymbol{s}\left(\boldsymbol{\theta}_t^m, x\right) - \boldsymbol{s}\left(\boldsymbol{\theta}_t^n, x\right) \right\|^2 \right] \right\| = 0 \ . \tag{7}$$

### 3.1 Comparison with Federated Learning and Distributed-SGD

Problem (3) can be viewed as a relaxation of the FL problem [3]. To see this, substitute $\boldsymbol{\theta}^m$ for $s\left(\boldsymbol{\theta}^m, x\right)$ and select a very large $\beta$. Then the regularization term in (3) forces the weights of all devices to coincide, as in FL and Distributed-SGD. Hence, the solution of (3) is $\min_{\boldsymbol{\theta}} \sum_{n=1}^{N} \mathcal{L}_n\left(\boldsymbol{\theta}\right)$, and devices communicate their weight vectors. The fact that D-Distillation, as opposed to FL, is not limited to this regime is the source for its communication reduction and its architecture-agnostic nature. Consequently, D-Distillation aims to minimize a different objective than FL, given in (3).

Matching weights is impossible in a network with heterogeneous devices that have different types of classifiers. This is the case for an IoT network, where many devices have proprietary classifiers (perhaps integrated chips) but still need software that can run distributed training across all of them. From an engineering point of view, one wants to include different DNN architectures to enjoy the diversity gain and mitigate overfitting, as done in ensemble learning [17].

Even in a homogeneous network, requiring $\boldsymbol{\theta}^n = \boldsymbol{\theta}^m$ for all $n, m$ might be excessive for non-convex environments. With DNNs, the loss landscape consists of plenty of local minima with very similar loss. Therefore, there is simply no need for all devices to agree on the same local minima, which only entails communicating the entire weight vectors. Our simulations confirm this intuition by showing that comparable performance gains to that of D-SGD can be achieved by only sharing soft-decisions.

| | Distributed-SGD [8–10] | Distributed Distillation |
|---|---|---|
| Data & Privacy | Private training data is generated locally and is never communicated to peers. | |
| Orchestration | No central system. Communication is in a peer-to-peer fashion. | |
| Communicate | DNN weights: scales with the size of the DNN, 23 million parameters for `ResNet-50` model | Soft-decisions: scales with $K$ classes, 10 for `CIFAR-10` dataset |
| Architecture | Identical classifier architectures required (e.g., same DNN model). | Arbitrary DNNs (or other classification models) allowed on any device. |

## 4 Convergence Analysis

In this section, we detail the proof strategy of Theorem 3 and discuss the main Lemmas, whose proofs are postponed to the Appendix. Our convergence analysis follows in two steps. First we establish that our algorithm reaches consensus on the network soft-decisions $\{\boldsymbol{z}^n\left(x\right)\}_n$ for all $x \in \mathcal{D}_r$. Then, we use a Lyapunov analysis to show that, given a consensus on the network soft-decisions, the gradients of the loss function in (3) all converge to zero. Our result follows by combining these two facts. The separation of these two converging processes is only for analysis purposes. In practice, they occur simultaneously and interact with each other.

Throughout the analysis, we let $z_t(x) = \left[z_t^1(x) \cdots z_t^N(x)\right] \in \mathbb{R}^{KN}$ be the concatenated vector of the network soft-decisions for $x \in \mathcal{D}_r$. These vectors are further stacked to form the vector of all network soft-decisions $z_t = [z_t(x_1) \cdots z_t(x_Q)] \in \mathbb{R}^{KNQ}$. We also define the vector of the network soft-decisions of device $n$ over all $x \in \mathcal{D}_r$, $z_t^n = [z_t^n(x_1) \cdots z_t^n(x_Q)] \in \mathbb{R}^{KQ}$.

We use the following filtration throughout the analysis, which summarizes the history of the algorithm:

$$\mathcal{F}_t = \sigma\left(\left\{\boldsymbol{\theta}_\tau, z_\tau, \mathcal{S}_{\tau-1}^r, \mathcal{S}_{\tau-1}^p \,|\, \tau \le t\right\}\right). \tag{8}$$

It will be useful to write (6) in matrix-vector form. To that end we define the stochastic gradient vector of the network soft-decisions, for all $x \in \mathcal{D}_r$, by $\boldsymbol{\varphi}_t(x) = \left[\boldsymbol{\varphi}_t^1(x) \cdots \boldsymbol{\varphi}_t^N(x)\right] \in \mathbb{R}^{KN}$, where

$$\boldsymbol{\varphi}_t^n(x) = 2\beta\left(z_t^n(x) - s\left(\boldsymbol{\theta}_t^n, x\right)\right). \tag{9}$$

Then

$$z_{t+1}(x) = \underbrace{(W \otimes I_K)}_{\tilde{W}} z_t(x) - \eta_t \boldsymbol{\varphi}_t(x) \tag{10}$$

where we defined $\tilde{W} = W \otimes I_K$ and $I_K$ is the identity matrix of size $K$. Note that since $W$ is doubly stochastic, so long as $z_t^n(x)$ is a probability vector, then $\tilde{W} z_t^n(x)$ is a probability vector (i.e., in $\Delta^K$). Since $s\left(\boldsymbol{\theta}_t^n, x\right) \in \Delta^K$, if $z_t^n(x) \in \Delta^K$, then the sum over the elements of $z_t^n(x) - s\left(\boldsymbol{\theta}_t^n, x\right)$ is zero. Hence, if we initialize $z_t^0(x) \in \Delta^K$ for all $n$ and $x$, then it follows by induction on $t$ that $z_t^n(x)$ is a probability vector for all $t$, $n$ and $x$. As a result, no projection is needed in our algorithm.

The next lemma shows that by running Algorithm 1, the devices reach an agreement on the network soft-decisions for each reference data point. Intuitively, the averaging step in (10), via the weighted multiplication by $W$, is aggressive enough that the gradient step with a decreasing step size cannot steer the dynamics away from the average. However, this average is not static, and the gradient steps of all devices move it towards convergence. This lemma justifies relaxing the problem from (3) to (4).

**Lemma 4.** *Let $\bar{z}_t(x) = \frac{1}{N} \sum_{n=1}^N z_t^n(x)$. Then $\sum_{n=1}^N \sum_{x \in \mathcal{D}_r} \|z_t^n(x) - \bar{z}_t(x)\|^2 \to 0$ with probability 1 as $t \to \infty$ and $\sum_{n=1}^N \sum_{x \in \mathcal{D}_r} \mathbb{E}\left\{\|z_t^n(x) - \bar{z}_t(x)\|^2\right\} \le C\eta_{t-1}^2$ for some constant $C > 0$.*

The second step of our convergence analysis is based on the following Lyapunov function:

$$\Phi(\boldsymbol{\theta}_t, \bar{z}_t) = \sum_{n=1}^N \mathcal{L}_n(\boldsymbol{\theta}_t^n) + \frac{\beta}{N} \sum_{x \in \mathcal{D}_r} \sum_{n=1}^N \|\bar{z}_t(x) - s\left(\boldsymbol{\theta}_t^n, x\right)\|^2 \tag{11}$$

for which

$$\nabla_{\boldsymbol{\theta}^n} \Phi(\boldsymbol{\theta}_t, \bar{z}_t) = \nabla\mathcal{L}_n(\boldsymbol{\theta}_t^n) + \frac{2\beta}{N} \sum_{x \in \mathcal{D}_r} \left(\nabla s\left(\boldsymbol{\theta}_t^n, x\right)\right)^T \left(s\left(\boldsymbol{\theta}_t^n, x\right) - \bar{z}_t(x)\right) \tag{12}$$

where $\nabla s\left(\boldsymbol{\theta}_t^n, x\right) \in \mathbb{R}^{K \times p_n}$, and

$$\nabla_{\bar{z}(x)} \Phi(\boldsymbol{\theta}_t, \bar{z}_t) = \frac{2\beta}{N} \sum_{n=1}^N \left(\bar{z}_t(x) - s\left(\boldsymbol{\theta}_t^n, x\right)\right). \tag{13}$$

The following lemma connects the algorithm updates in (5) and (6) to the gradients of the Lyapunov function in (11). As opposed to the usual SGD analysis, here the stochastic gradients are biased. This bias arises since the stochastic gradients are computed with respect to the local network soft-decisions $z^n(x)$ and not the average network soft-decisions $\bar{z}(x)$. However, the average of the stochastic gradient with respect to $z^n(x)$ over all devices $n$ is unbiased. The bias of the stochastic gradient with respect to $\boldsymbol{\theta}^n$ vanishes as the consensus on $\{z_t^n\}_n$ is reached, which is guaranteed by Lemma 4.

**Lemma 5.** *Let $\mathcal{S}_t^p, \mathcal{S}_t^r$ be the random subsets used in (5). Define device $n$'s weight gradient at time $t$*

$$\boldsymbol{g}_t^n \triangleq \frac{M_n}{|\mathcal{S}_t^p|} \sum_{\tilde{x} \in \mathcal{S}_t^p} \nabla\mathcal{L}\left(s\left(\boldsymbol{\theta}_t^n, \tilde{x}\right), \boldsymbol{y}(\tilde{x})\right) + \frac{2\beta Q}{N|\mathcal{S}_t^r|} \sum_{x \in \mathcal{S}_t^r} \left(\nabla s\left(\boldsymbol{\theta}_t^n, x\right)\right)^T \left(s\left(\boldsymbol{\theta}_t^n, x\right) - z_t^n(x)\right) \tag{14}$$

*and for each $x \in \mathcal{D}_r$ let the network soft-decision gradient be*

$$\boldsymbol{\varphi}_t^n(x) = 2\beta\left(z_t^n(x) - s\left(\boldsymbol{\theta}_t^n, x\right)\right). \tag{15}$$

*Then with probability 1, for every $n$ and every $x \in \mathcal{D}_r$:*

1. $\mathbb{E}\left\{\boldsymbol{g}_t^n \mid \mathcal{F}_t\right\} = \nabla_{\boldsymbol{\theta}^n}\Phi\left(\boldsymbol{\theta}_t, \boldsymbol{z}_t^n\right)$ *and* $\|\boldsymbol{g}_t^n\|^2 \le M_n^2 L_a^2 + \frac{4\sqrt{2}\beta}{N}M_n L_a L_s Q + \frac{8\beta^2}{N^2}L_s^2 Q^2.$

2. $\mathbb{E}\left\{\frac{1}{N}\sum_{n=1}^{N}\boldsymbol{\varphi}_t^n\left(x\right) \mid \mathcal{F}_t\right\} = \nabla_{\bar{\boldsymbol{z}}(x)}\Phi\left(\boldsymbol{\theta}_t, \bar{\boldsymbol{z}}_t\right)$ *and* $\|\boldsymbol{\varphi}_t^n\left(x\right)\|^2 \le 8\beta^2.$

Finally we show that our Lyapunov function has Lipschitz continuous gradients, by showing that the second term in (11) also has Lipschitz continuous gradients.

**Lemma 6.** *The function* $h\left(\boldsymbol{\theta}, \bar{\boldsymbol{z}}\right) = \sum_{x\in\mathcal{D}_r}\sum_{n=1}^{N}\|\bar{\boldsymbol{z}}\left(x\right) - s\left(\boldsymbol{\theta}^n, x\right)\|^2$ *has Lipschitz continuous gradients with some constant* $L_h > 0$ *(specified in the proof).*

## 5 Simulation Results

We conduct DNN simulations to evaluate the performance of Distributed Distillation (D-Distillation) compared to two baselines: Distributed-SGD (D-SGD) and Silo-SGD (where each device trains its DNN with only its private data and no communication). These simulations serve as a proof-of-concept for the new D-Distillation archetype algorithm and evaluate the gain of distilling the knowledge from the whole network as our main result guarantees. We selected the best hyperparameters for each algorithm from a limited search as detailed in Appendix 12.1.

Our simulations validate that D-Distillation transfers knowledge across devices, which results in significant performance gains over Silo-SGD. Moreover, Fig. 2 shows that D-Distribution obtains comparable accuracy to D-SGD for a 16 device network, but it reaches $95.8\%$ with a $46\times$ communication reduction versus D-SGD. As more devices are added to the network, the total private data in the network, which each device distills, increases. Section 12.2.1 in the Appendix confirms this intuition and shows that D-Distillation's performance generally improves with the number of devices.

Our experiment in Fig. 3 shows the accuracy gain D-Distillation obtains when the communication network includes devices with different local models (e.g., as in an IoT network). In practice, this means that D-Distillation can outperform D-SGD by allowing more devices (and thereby more private data) into the training process. Even by joining just two subsets with different models, our method outperforms D-SGD which is forced to run on each subset of devices separately. D-Distillation further increases its performance over D-SGD when there are four subsets with different models.

D-Distillation achieves substantial communication reduction compared to D-SGD by changing the object that is being communicated and targeting a different type of stationary points. On top of this novel source for communication reduction, one can apply state-of-the-art communication reduction schemes such as [3, 7, 28] to the network soft-decisions in D-Distillation with little effort. In Fig. 4, we demonstrate that D-Distillation can easily achieve an *additional* $90\times$ *communication reduction with no accuracy loss for a total of* $9{,}370\times$ *communication reduction.*

Our implementation of D-Distillation does not utilize all available degrees of freedom, so the full potential of D-Distillation has yet to be unleashed. This has the potential of closing the 8 points accuracy gap to D-SGD of Fig. 4, or even surpassing D-SGD. For instance, we did not optimize

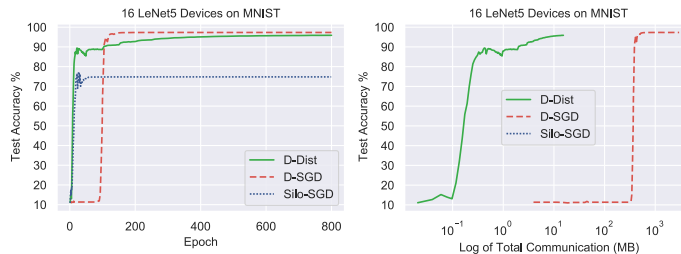

Figure 2: Comparison of Silo-SGD, D-SGD, and D-Distillation mean test accuracy by training epochs (left) and total communication (right) over 16 devices on a random communication graph instance of maximum degree 3, training `LeNet-5` on `MNIST` [40]. D-Distillation uses network batch size $b = 32$. The reference dataset was composed of a random $40\%$ of the `MNIST` training data and the remaining training data was evenly distributed at random to be devices' private training data.

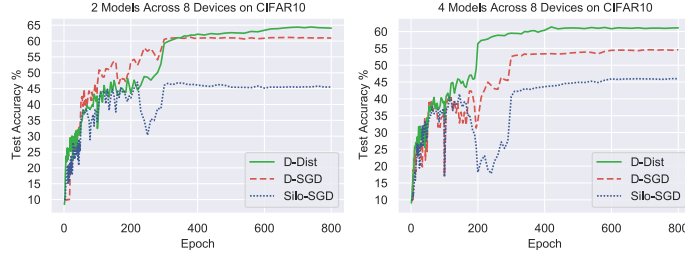

Figure 3: We simulate a network of 8 devices where different devices have different classification models. On the left half the devices are training a LeNet-5 and half a ResNet-8 on `CIFAR10` [41]. On the right there are two devices training each of the four models: LeNet-5 [40], ResNet-2, ResNet-8, and ResNet-14 [42]. D-Distillation learns across all 8 devices with different models to outperform the accuracy of D-SGD while achieving a 540x lower communication overhead per device.

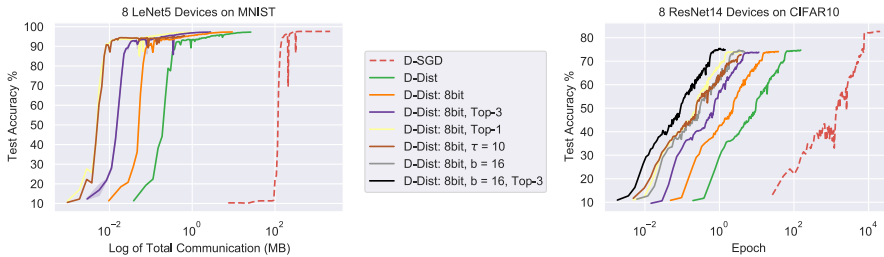

Figure 4: We demonstrate how D-Distillation can be combined with the existing communication reduction schemes from [3,7,28] of only communicating every $\tau$ gradient steps ($\tau = 10$), quantization (8 bit) and Top-K compression (i.e., keep the largest $K = 3$ elements in each soft decision). Furthermore, we show that we can reduce the network batch size, $b$, communicated in each round of D-Distillation to further reduce communication. The default network batch size used on `MNIST` (left) is $b = 32$ and on `CIFAR10` (right) is $b = 128$, such that they both match the local SGD batch size. In all of these methods there is little to no accuracy loss and sometimes even a gain.

the unlabeled reference dataset, which can be handcrafted and designed for improved performance. Additional results and discussion are in Appendix 12.

## 6   Conclusions

We presented a new distributed training paradigm where the devices' classifiers learn from each other by communicating soft-decision variables on a reference dataset. This way, devices never send their private data. Assuming smooth but non-convex loss functions, we prove that our algorithm guarantees that with probability 1, the gradients of all distillation loss functions of all devices converge to zero, so devices distill the knowledge of all other devices in the communication network, despite their limited local connections. Hence, our algorithm is a distributed version of the distillation algorithms [18–21, 23] for any strongly connected, directed communication graph.

Compared to FL algorithms that communicate model weights, our approach reduces the communication overhead significantly and allows for devices with different classifier architectures (e.g., proprietary models) to participate in the distributed training. Simulations show that a naive implementation of our algorithm on a network of 16 devices achieves comparable accuracy while reducing the communication overhead by $46\times$ compared to D-SGD without sacrificing performance. Implementing common communication reduction schemes on top of our novel method results in a total communication reduction of $9,370\times$ compared to D-SGD. Scaling up to larger communication networks would make the communication reduction even more significant. The encouraging results of our prototype algorithm call for experimental work to explore the full potential of our new approach.

## Broader Impact

This work deals with combining the training efforts of edge devices to obtain better machine learning models on each device. It scales up machine learning, making ML applications available on many more devices and accessible to many more people with all the positive and the negative effects that may ensue. At the same time, it maintains the setting where users keep their data on their device which allows for increased privacy and prevents the monopolistic aggregation of user data by large companies. A well functioning smart home IoT device that never records your conversations to the company's servers should be an important milestone. By addressing the decentralized regime, Distributed Distillation further removes the dependency on a central authority, which contributes to making machine learning more open and democratic in nature. With no central server required to process any data or parameters, this work lowers the barrier to entry to deploy such an algorithm. On the negative side, the issue of bias in machine learning potentially increases in a decentralized system where it is harder to evaluate the decision outcomes and all countermeasures must also be implemented in a decentralized fashion.

## Acknowledgments and Disclosure of Funding

This work was funded by a Stanford SystemX Seed Grant and by the Koret Foundation grant for Smart Cities and Digital Living.

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
