[Supplementary Material]

# 7 Proof of Lemma 4

*Proof.* Let $b = |\mathcal{S}_t^r|$, which is constant for all $t$. Define the total disagreement error as

$$\phi(z_t) \triangleq \frac{b}{Q} \sum_{x \in \mathcal{D}_r} \left\| \left(I - \tilde{W}\right) z_t(x) \right\|^2. \tag{16}$$

Let $\lambda_{N-1} = \lambda_{N-1}\left(W^T W\right)$. Using (10) we can write

$$\phi(z_{t+1}) \underset{(a)}{=} \mathbb{E}\left\{ \sum_{x \in \mathcal{S}_t^r} \left\| \left(I - \tilde{W}\right) z_{t+1}(x) \right\|^2 \mid \mathcal{F}_t \right\}.$$

$$= \mathbb{E}\left\{ \sum_{x \in \mathcal{S}_t^r} \left\| \left(I - \tilde{W}\right)\left(\tilde{W} z_t(x) - \eta_t \varphi_t(x)\right) \right\|^2 \mid \mathcal{F}_t \right\}$$

$$\underset{(b)}{\leq} \mathbb{E}\left\{ \sum_{x \in \mathcal{S}_t^r} \underbrace{\left\| \tilde{W}\left(I - \tilde{W}\right) z_t(x) \right\|^2}_{A^2} \mid \mathcal{F}_t \right\}$$

$$+ 2\eta_t \mathbb{E}\left\{ \sum_{x \in \mathcal{S}_t^r} \underbrace{\left\| \tilde{W}\left(I - \tilde{W}\right) z_t(x) \right\|}_{A} \underbrace{\left\| \left(I - \tilde{W}\right) \varphi_t(x) \right\|}_{B} \mid \mathcal{F}_t \right\}$$

$$+ \eta_t^2 \mathbb{E}\left\{ \sum_{x \in \mathcal{S}_t^r} \underbrace{\left\| \left(I - \tilde{W}\right) \varphi_t(x) \right\|^2}_{B^2} \mid \mathcal{F}_t \right\}$$

$$\underset{(c)}{\leq} \mathbb{E}\left\{ \lambda_{N-1} \sum_{x \in \mathcal{S}_t^r} \left\| \left(I - \tilde{W}\right) z_t(x) \right\|^2 \mid \mathcal{F}_t \right\}$$

$$+ 4\sqrt{\lambda_{N-1} D_0}\, \eta_t \mathbb{E}\left\{ \sum_{x \in \mathcal{S}_t^r} \left\| \left(I - \tilde{W}\right) z_t(x) \right\| \mid \mathcal{F}_t \right\} + 4b\eta_t^2 D_0$$

$$= \lambda_{N-1} \phi(z_t) + 4\sqrt{\lambda_{N-1} D_0}\, \eta_t \frac{b}{Q} \sum_{x \in \mathcal{D}_r} \left\| \left(I - \tilde{W}\right) z_t(x) \right\| + 4b\eta_t^2 D_0$$

$$\underset{(d)}{\leq} \lambda_{N-1} \phi(z_t) + 4\eta_t \sqrt{\lambda_{N-1} D_0 b}\sqrt{\phi(z_t)} + 4b\eta_t^2 D_0 \tag{17}$$

where (a) follows since given $\mathcal{F}_t$, the only randomness left is in $\mathcal{S}_t^r$, which includes each $x \in \mathcal{D}_r$ with probability $\frac{b}{Q}$. Inequality (b) is Cauchy-Schwartz and $\tilde{W}\left(I - \tilde{W}\right) = \left(I - \tilde{W}\right)\tilde{W}$. Inequality (c) uses part 3 of Lemma 7 on the A terms in (17) and part 1 of Lemma 7 on the B terms in (17). It also uses $\|\varphi_t(x)\|^2 \leq D_0$ where $D_0$ is a constant specified in Lemma 5. Inequality (d) uses $\left(\sum_{x \in \mathcal{D}_r} \left\| \left(I - \tilde{W}\right) z_t(x) \right\|\right)^2 \leq Q \sum_{x \in \mathcal{D}_r} \left\| \left(I - \tilde{W}\right) z_t(x) \right\|^2$ which follows from Lemma 8 for scalars.

Taking an expectation on both sides of (17) we obtain

$$\mathbb{E}\left\{\phi(z_{t+1})\right\} \leq \lambda_{N-1}\mathbb{E}\left\{\phi(z_t)\right\} + 4\eta_t \sqrt{\lambda_{N-1} D_0 b}\,\mathbb{E}\left\{\sqrt{\phi(z_t)}\right\} + 4bD_0\eta_t^2$$

$$\leq \lambda_{N-1}\mathbb{E}\left\{\phi(z_t)\right\} + 4\eta_t \sqrt{\lambda_{N-1} D_0 b \mathbb{E}\left\{\phi(z_t)\right\}} + 4bD_0\eta_t^2. \tag{18}$$

We next prove by induction that for all $t \geq 1$

$$\mathbb{E}\left\{\phi(z_{t+1})\right\} \leq A\eta_t^2 \tag{19}$$

for some constant $A > 0$ that we specify next.

Choose a constant $A_0$ such that

$$\frac{A_0 - 8\sqrt{\lambda_{N-1} D_0 b A_0} - 4bD_0}{A_0 \lambda_{N-1}} > 1 \implies (1 - \lambda_{N-1}) A_0 - 8\sqrt{\lambda_{N-1} b D_0}\sqrt{A_0} - 4bD_0 > 0$$

$$\implies \sqrt{A_0} > 2\sqrt{bD_0}\frac{2\sqrt{\lambda_{N-1}} + \sqrt{3\lambda_{N-1} + 1}}{1 - \lambda_{N-1}} \tag{20}$$

where we also used that $\lambda_{N-1} < 1$, from Lemma 7.

Define $T_0$ to be large enough such that $\eta_{t-1} \le 2\eta_t$ and $\eta_{t-1} \le \sqrt{\frac{A_0 - 8\sqrt{\lambda_{N-1} D_0 b A_0} - 4bD_0}{A_0 \lambda_{N-1}}}\eta_t$ for all $t \ge T_0$. Such large enough $T_0$ exists since $\lim_{t\to\infty} \frac{\eta_{t+1}}{\eta_t}$ exists, and it must be equal to one otherwise the ratio test [43] will either say that $\sum_{t=1}^{\infty} \eta_t = \infty$ converges or that $\sum_{t=1}^{\infty} \eta_t^2 < \infty$ does not converge.

Then we pick $A = \max\left\{\max_{1 < t \le T_0} \frac{1}{\eta_{t-1}^2}\mathbb{E}\{\phi(z_t)\}, A_0\right\}$ which is constant with respect to $t$. Hence for all $t \le T_0$ the induction hypothesis (19) holds.

Now assume that (19) holds for some $t \ge T_0$. Plugging this in (18) completes the induction:

$$\mathbb{E}\{\phi(z_{t+1})\} \le A\lambda_{N-1}\eta_{t-1}^2 + 4\sqrt{\lambda_{N-1} D_0 b A}\eta_t\eta_{t-1} + 4\eta_t^2 bD_0$$

$$\underset{(a)}{\le} \left(A - 8\sqrt{\lambda_{N-1} D_0 bA} - 4bD_0\right)\eta_t^2 + 8\sqrt{\lambda_{N-1} D_0 bA}\eta_t^2 + 4\eta_t^2 bD_0$$

$$\le A\eta_t^2 \tag{21}$$

where (a) follows for all $t \ge T_0$, since then $A \ge A_0$, so $\eta_{t-1} \le \sqrt{\frac{A - 8\sqrt{\lambda_{N-1} D_0 bA} - 4bD_0}{A\lambda_{N-1}}}\eta_t$ and also $\eta_{t-1} \le 2\eta_t$.

Using (19), we obtain that for any $\varepsilon > 0$

$$\sum_{t=1}^{\infty} \Pr(\phi(z_t) \ge \varepsilon) \le 1 + \sum_{t=2}^{\infty} \frac{\mathbb{E}\{\phi(z_t)\}}{\varepsilon} \le 1 + \frac{A}{\varepsilon}\sum_{t=1}^{\infty} \eta_t^2 < \infty \tag{22}$$

so by the Borel-Cantelli Lemma, for almost all $\omega \in \Omega$, where $\Omega$ is the probability space, there exists $T(\omega)$ such that $\phi(z_t) \le \varepsilon$ for every $t > T(\omega)$. Since $\phi(z_t)$ is non-negative, this means that $\lim_{t\to\infty} \phi(z_t) = 0$ with probability 1. Let $\lambda_2 = \lambda_2\left((I - W)^T (I - W)\right)$. Then

$$\phi(z_t) = \frac{b}{Q}\sum_{x \in \mathcal{D}_r} \left\|\left(I - \tilde{W}\right) z_t(x)\right\|^2 \underset{(a)}{\ge}$$

$$\lambda_2 \frac{b}{Q}\sum_{x \in \mathcal{D}_r} \|z_t(x) - \mathbf{1}_N \otimes \bar{z}_t(x)\|^2 = \lambda_2 \frac{b}{Q}\sum_{x \in \mathcal{D}_r}\sum_{n=1}^{N} \|z_t^n(x) - \bar{z}_t(x)\|^2 \tag{23}$$

where (a) follows from part 2 of Lemma 7. We conclude that since $\phi(z_t) \to 0$ with probability 1 and $\lambda_2 > 0$ then $\sum_{x \in \mathcal{D}_r}\sum_{n=1}^{N} \|z_t^n(x) - \bar{z}_t(x)\|^2 \to 0$ with probability 1. Moreover, (19) and (23) show that $\sum_{n=1}^{N}\sum_{x \in \mathcal{D}_r} \mathbb{E}\left\{\|z_t^n(x) - \bar{z}_t(x)\|^2\right\} \le C\eta_{t-1}^2$ for some constant $C > 0$. $\square$

# 8 Proof of Lemma 5

*Proof.* Since $\boldsymbol{\theta}_t^n, \boldsymbol{z}_t^n$ are $\mathcal{F}_t$ measurable, we have

$$\mathbb{E}\{\boldsymbol{g}_t^n \mid \mathcal{F}_t\} =$$
$$\sum_{\tilde{x} \in \mathcal{D}_n} \nabla \mathcal{L}\left(s\left(\boldsymbol{\theta}_t^n, \tilde{x}\right), \boldsymbol{y}\left(\tilde{x}\right)\right) + \frac{2\beta}{N} \sum_{x \in \mathcal{D}_r} \left(\nabla s\left(\boldsymbol{\theta}_t^n, x\right)\right)^T \left(s\left(\boldsymbol{\theta}_t^n, x\right) - \boldsymbol{z}_t^n\left(x\right)\right) = \nabla_{\theta^n} \Phi\left(\boldsymbol{\theta}_t, \boldsymbol{z}_t^n\right).$$

$$(24)$$

Moreover,

$$\|\boldsymbol{g}_t^n\|^2 \underset{(a)}{\leq}$$

$$\frac{M_n^2}{|\mathcal{S}_t^p|^2} \left\|\sum_{\tilde{x} \in \mathcal{S}_t^p} \nabla \mathcal{L}\left(s\left(\boldsymbol{\theta}_t^n, \tilde{x}\right), \boldsymbol{y}\left(\tilde{x}\right)\right)\right\|^2 + \frac{4\beta^2 Q^2}{N^2 |\mathcal{S}_t^r|^2} \left\|\sum_{x \in \mathcal{S}_t^r} \left(\nabla s\left(\boldsymbol{\theta}_t^n, x\right)\right)^T \left(s\left(\boldsymbol{\theta}_t^n, x\right) - \boldsymbol{z}_t^n\left(x\right)\right)\right\|^2 +$$

$$\frac{4\beta}{N} \frac{M_n Q}{|\mathcal{S}_t^p| |\mathcal{S}_t^r|} \left\|\sum_{\tilde{x} \in \mathcal{S}_t^p} \nabla \mathcal{L}\left(s\left(\boldsymbol{\theta}_t^n, \tilde{x}\right), \boldsymbol{y}\left(\tilde{x}\right)\right)\right\| \left\|\sum_{x \in \mathcal{S}_t^r} \left(\nabla s\left(\boldsymbol{\theta}_t^n, x\right)\right)^T \left(s\left(\boldsymbol{\theta}_t^n, x\right) - \boldsymbol{z}_t^n\left(x\right)\right)\right\| \underset{(b)}{\leq}$$

$$\frac{M_n^2}{|\mathcal{S}_t^p|} \sum_{\tilde{x} \in \mathcal{S}_t^p} \|\nabla \mathcal{L}\left(s\left(\boldsymbol{\theta}_t^n, \tilde{x}\right), \boldsymbol{y}\left(\tilde{x}\right)\right)\|^2 + \frac{4\beta^2 Q^2}{N^2 |\mathcal{S}_t^r|} \sum_{x \in \mathcal{S}_t^r} \left\|\left(\nabla s\left(\boldsymbol{\theta}_t^n, x\right)\right)^T \left(s\left(\boldsymbol{\theta}_t^n, x\right) - \boldsymbol{z}_t^n\left(x\right)\right)\right\|^2 +$$

$$\frac{4\beta}{N} \frac{M_n Q}{|\mathcal{S}_t^p| |\mathcal{S}_t^r|} \sum_{\tilde{x} \in \mathcal{S}_t^p} \|\nabla \mathcal{L}\left(s\left(\boldsymbol{\theta}_t^n, \tilde{x}\right), \boldsymbol{y}\left(\tilde{x}\right)\right)\| \sum_{x \in \mathcal{S}_t^r} \left\|\left(\nabla s\left(\boldsymbol{\theta}_t^n, x\right)\right)^T \left(s\left(\boldsymbol{\theta}_t^n, x\right) - \boldsymbol{z}_t^n\left(x\right)\right)\right\| \underset{(c)}{\leq}$$

$$M_n^2 L_a^2 + \frac{4\sqrt{2}\beta}{N} M_n L_a L_s Q + \frac{8\beta^2}{N^2} L_s^2 Q^2 \quad (25)$$

where (a) uses Cauchy-Schwarz, (b) uses the triangle inequality and Lemma 8. Inequality (c) uses the Lipschitz continuity of $\mathcal{L}\left(s\left(\boldsymbol{\theta}_t^n, \tilde{x}\right), \boldsymbol{y}\left(\tilde{x}\right)\right)$ (see Lemma 10) with some parameter $L_a$. Inequality (c) also uses that $s\left(\boldsymbol{\theta}_t^n, x\right) - \boldsymbol{z}_t^n\left(x\right)$ is a difference between two probability vectors, after using that for all $\boldsymbol{y}$

$$\|\nabla s\left(\boldsymbol{\theta}_t^n, x\right) \boldsymbol{y}\|^2 \underset{(a)}{\leq} \|\nabla s\left(\boldsymbol{\theta}_t^n, x\right)\|^2 \|\boldsymbol{y}\|^2 \underset{(b)}{\leq} L_s^2 \|\boldsymbol{y}\|^2 \quad (26)$$

where (a) follows from the submultiplicativy of the spectral norm, and (b) since $s\left(\boldsymbol{\theta}_t^n, x\right)$ is Lipschitz continuous with parameter $L_s$ (and continuously differentiable since the gradients are Lipschitz continuous).

For the second part of the lemma, we readily obtain that for each $x \in \mathcal{D}_r$

$$\mathbb{E}\left\{\frac{1}{N} \sum_{n=1}^N \boldsymbol{\varphi}_t^n\left(x\right) \mid \mathcal{F}_t\right\} = \frac{2\beta}{N} \sum_{n=1}^N \left(\boldsymbol{z}_t^n\left(x\right) - s\left(\boldsymbol{\theta}_t^n, x\right)\right) = \nabla_{\bar{z}(x)} \Phi\left(\boldsymbol{\theta}_t, \bar{\boldsymbol{z}}_t\right) \quad (27)$$

and that for every $n$ and $x \in \mathcal{D}_r$

$$\|\boldsymbol{\varphi}_t^n\left(x\right)\|^2 = 4\beta^2 \|s\left(\boldsymbol{\theta}_t^n, x\right) - \boldsymbol{z}_t^n\left(x\right)\|^2 \underset{(a)}{\leq} 8\beta^2 \quad (28)$$

where (a) uses that $s\left(\boldsymbol{\theta}_t^n, x\right) - \boldsymbol{z}_t^n\left(x\right)$ is the difference between two probability vectors. $\square$

# 9 Proof of Lemma 6

*Proof.* For each $n$

$$\nabla_{\boldsymbol{\theta}^n} h\left(\boldsymbol{\theta}, \bar{\boldsymbol{z}}\right) = 2 \sum_{x \in \mathcal{D}_r} \left(\nabla s\left(\boldsymbol{\theta}^n, x\right)\right)^T \left(s\left(\boldsymbol{\theta}^n, x\right) - \bar{\boldsymbol{z}}\left(x\right)\right) \quad (29)$$

and

$$\nabla_{\bar{z}(x)} h\left(\boldsymbol{\theta}, \bar{\boldsymbol{z}}\right) = 2 \sum_{n=1}^N \left(\bar{\boldsymbol{z}}\left(x\right) - s\left(\boldsymbol{\theta}^n, x\right)\right). \quad (30)$$

Then

$$\left\| (\nabla s\,(\boldsymbol{\theta}^n, x))^T \,(s\,(\boldsymbol{\theta}^n, x) - \bar{\boldsymbol{z}}\,(x)) - (\nabla s\,(\boldsymbol{\nu}^n, x))^T \,(s\,(\boldsymbol{\nu}^n, x) - \bar{\boldsymbol{y}}\,(x)) \right\|^2 \underset{(a)}{\leq}$$

$$3 \left\| (\nabla s\,(\boldsymbol{\theta}^n, x))^T \,(s\,(\boldsymbol{\theta}^n, x) - \bar{\boldsymbol{z}}\,(x)) - (\nabla s\,(\boldsymbol{\theta}^n, x))^T \,(s\,(\boldsymbol{\nu}^n, x) - \bar{\boldsymbol{z}}\,(x)) \right\|^2 +$$

$$3 \left\| (\nabla s\,(\boldsymbol{\theta}^n, x))^T \,(s\,(\boldsymbol{\nu}^n, x) - \bar{\boldsymbol{z}}\,(x)) - (\nabla s\,(\boldsymbol{\theta}^n, x))^T \,(s\,(\boldsymbol{\nu}^n, x) - \bar{\boldsymbol{y}}\,(x)) \right\|^2 +$$

$$3 \left\| (\nabla s\,(\boldsymbol{\theta}^n, x))^T \,(s\,(\boldsymbol{\nu}^n, x) - \bar{\boldsymbol{y}}\,(x)) - (\nabla s\,(\boldsymbol{\nu}^n, x))^T \,(s\,(\boldsymbol{\nu}^n, x) - \bar{\boldsymbol{y}}\,(x)) \right\|^2 \underset{(b)}{\leq}$$

$$3 L_s^2 \left\| s\,(\boldsymbol{\theta}^n, x) - s\,(\boldsymbol{\nu}^n, x) \right\|^2 + 3 L_s^2 \left\| \bar{\boldsymbol{z}}\,(x) - \bar{\boldsymbol{y}}\,(x) \right\|^2 +$$

$$3 \left\| (\nabla s\,(\boldsymbol{\theta}^n, x) - \nabla s\,(\boldsymbol{\nu}^n, x))^T \,(s\,(\boldsymbol{\nu}^n, x) - \bar{\boldsymbol{y}}\,(x)) \right\|^2 \underset{(c)}{\leq}$$

$$3 \left( L_s^4 + 2 L_g^2 \right) \left\| \boldsymbol{\theta}^n - \boldsymbol{\nu}^n \right\|^2 + 3 L_s^2 \left\| \bar{\boldsymbol{z}}\,(x) - \bar{\boldsymbol{y}}\,(x) \right\|^2 \quad (31)$$

where (a) uses Lemma 8 for a sum of three terms and (b) follows since $\left\| \nabla s\,(\boldsymbol{\theta}^n, x) \right\|^2 \leq L_s^2$. Inequality (c) follows from the Lipschitz continuity of $s\,(\boldsymbol{\theta}^n, x)$ and $\nabla s\,(\boldsymbol{\theta}^n, x)$, with parameters $L_s$ and $L_g$, and since for the spectral norm of $\nabla s\,(\boldsymbol{\theta}^n, x) - \nabla s\,(\boldsymbol{\nu}^n, x)$ we have

$$\left\| (\nabla s\,(\boldsymbol{\theta}^n, x) - \nabla s\,(\boldsymbol{\nu}^n, x))^T \,(s\,(\boldsymbol{\nu}^n, x) - \bar{\boldsymbol{y}}\,(x)) \right\|^2 \leq$$

$$\left\| \nabla s\,(\boldsymbol{\theta}^n, x) - \nabla s\,(\boldsymbol{\nu}^n, x) \right\|^2 \left\| s\,(\boldsymbol{\nu}^n, x) - \bar{\boldsymbol{y}}\,(x) \right\|^2 \leq 2 L_g^2 \left\| \boldsymbol{\theta}^n - \boldsymbol{\nu}^n \right\|^2. \quad (32)$$

where $\left\| s\,(\boldsymbol{\nu}^n, x) - \bar{\boldsymbol{y}}\,(x) \right\|^2 \leq 2$ as a difference of probability vectors.

Hence by using Lemma 8 on (29)

$$\left\| \nabla_{\boldsymbol{\theta}^n} h\,(\boldsymbol{\theta}, \bar{\boldsymbol{z}}) - \nabla_{\boldsymbol{\nu}^n} h\,(\boldsymbol{\nu}, \bar{\boldsymbol{y}}) \right\|^2 \leq 12 Q^2 \left( L_s^4 + 2 L_g^2 \right) \left\| \boldsymbol{\theta}^n - \boldsymbol{\nu}^n \right\|^2 + 12 Q L_s^2 \left\| \bar{\boldsymbol{z}} - \bar{\boldsymbol{y}} \right\|^2. \quad (33)$$

Next we obtain

$$\left\| \nabla_{\bar{\boldsymbol{z}}} h\,(\boldsymbol{\theta}, \bar{\boldsymbol{z}}) - \nabla_{\bar{\boldsymbol{y}}} h\,(\boldsymbol{\nu}, \bar{\boldsymbol{y}}) \right\|^2 = 4 \sum_{x \in \mathcal{D}_r} \left\| \sum_{n=1}^{N} (\bar{\boldsymbol{z}}\,(x) - s\,(\boldsymbol{\theta}^n, x)) - (\bar{\boldsymbol{y}}\,(x) - s\,(\boldsymbol{\nu}^n, x)) \right\|^2 \underset{(a)}{\leq}$$

$$8N \sum_{x \in \mathcal{D}_r} \left( \sum_{n=1}^{N} \left\| s\,(\boldsymbol{\theta}^n, x) - s\,(\boldsymbol{\nu}^n, x) \right\|^2 + \sum_{n=1}^{N} \left\| \bar{\boldsymbol{z}}\,(x) - \bar{\boldsymbol{y}}\,(x) \right\|^2 \right) \leq$$

$$8 L_s^2 N Q \left\| \boldsymbol{\theta} - \boldsymbol{\nu} \right\|^2 + 8N^2 \left\| \bar{\boldsymbol{z}} - \bar{\boldsymbol{y}} \right\|^2 \quad (34)$$

where (a) uses Lemma 8 with $2N$ terms and (b) uses the $L_s$-Lipschitz continuity of $s\,(\boldsymbol{\theta}^n, x)$. We conclude by gathering all the components of the concatenated vectors $(\boldsymbol{\theta}, \boldsymbol{z})$ and $(\boldsymbol{\nu}, \boldsymbol{y})$ together:

$$\left\| \nabla h\,(\boldsymbol{\theta}, \bar{\boldsymbol{z}}) - \nabla h\,(\boldsymbol{\nu}, \bar{\boldsymbol{y}}) \right\|^2 =$$

$$\sum_{n=1}^{N} \left\| \nabla_{\boldsymbol{\theta}^n} h\,(\boldsymbol{\theta}, \bar{\boldsymbol{z}}) - \nabla_{\boldsymbol{\nu}^n} h\,(\boldsymbol{\nu}, \bar{\boldsymbol{y}}) \right\|^2 + \left\| \nabla_{\bar{\boldsymbol{z}}} h\,(\boldsymbol{\theta}, \bar{\boldsymbol{z}}) - \nabla_{\bar{\boldsymbol{y}}} h\,(\boldsymbol{\nu}, \bar{\boldsymbol{y}}) \right\|^2 \leq$$

$$\left( 12 Q^2 L_s^4 + 24 Q^2 L_g^2 + 8 L_s^2 N Q \right) \left\| \boldsymbol{\theta} - \boldsymbol{\nu} \right\|^2 + \left( 12 N Q L_s^2 + 8N^2 \right) \left\| \bar{\boldsymbol{z}} - \bar{\boldsymbol{y}} \right\|^2 \leq$$

$$\max \left\{ 12 Q^2 L_s^4 + 24 Q^2 L_g^2 + 8 L_s^2 N Q, 12 N Q L_s^2 + 8N^2 \right\} \left\| (\boldsymbol{\theta}, \bar{\boldsymbol{z}}) - (\boldsymbol{\nu}, \bar{\boldsymbol{y}}) \right\|^2. \quad (35)$$

$$\square$$

## 10 Proof of Theorem 3

*Proof.* For all $x \in \mathcal{D}_r$, define $\bar{\varphi}_t (x) = \frac{1}{N} \sum_{n=1}^{N} \varphi_t^n (x)$. Let $\boldsymbol{a} = \left[ \frac{1}{N}, \ldots, \frac{1}{N} \right]$ be the averaging vector of size $N$, such that $\bar{\boldsymbol{z}}_t (x) = (\boldsymbol{a} \otimes I_K) \boldsymbol{z}_t (x)$. Then from (10) we have

$$\bar{\boldsymbol{z}}_{t+1} (x) = \frac{1}{N} \sum_{n=1}^{N} \boldsymbol{z}_{t+1}^n (x) = (\boldsymbol{a} \otimes I_K) (W \otimes I_K) \boldsymbol{z}_t (x) - \eta_t \frac{1}{N} \sum_{n=1}^{N} \varphi_t^n (x) \underset{(a)}{=}$$

$$(\boldsymbol{a} W \otimes I_K) \boldsymbol{z}_t (x) - \eta_t \frac{1}{N} \sum_{n=1}^{N} \varphi_t^n (x) \underset{(b)}{=}$$

$$(\boldsymbol{a} \otimes I_K) \boldsymbol{z}_t (x) - \eta_t \frac{1}{N} \sum_{n=1}^{N} \varphi_t^n (x) = \frac{1}{N} \sum_{n=1}^{N} \boldsymbol{z}_t^n (x) - \eta_t \frac{1}{N} \sum_{n=1}^{N} \varphi_t^n (x) \quad (36)$$

where (a) follows since $(A \otimes B)(C \otimes D) = AC \otimes BD$ and (b) since $\boldsymbol{a} W = \boldsymbol{a}$, using that $W$ is doubly stochastic.

From Lemma 5 we have for all $x \in \mathcal{D}_r$

$$\mathbb{E} \{ \bar{\varphi}_t (x) \mid \mathcal{F}_t \} = \frac{1}{N} \sum_{n=1}^{N} \mathbb{E} \{ \varphi_t^n (x) \mid \mathcal{F}_t \} = \nabla_{\bar{\boldsymbol{z}}(x)} \Phi (\boldsymbol{\theta}_t, \bar{\boldsymbol{z}}_t). \quad (37)$$

Lemma 5 also gives,

$$\mathbb{E} \{ \boldsymbol{g}_t^n \mid \mathcal{F}_t \} = \nabla_{\boldsymbol{\theta}^n} \Phi (\boldsymbol{\theta}_t, \boldsymbol{z}_t^n) =$$
$$\nabla_{\boldsymbol{\theta}^n} \Phi (\boldsymbol{\theta}_t, \bar{\boldsymbol{z}}_t) + (\nabla_{\boldsymbol{\theta}^n} \Phi (\boldsymbol{\theta}_t, \boldsymbol{z}_t^n) - \nabla_{\boldsymbol{\theta}^n} \Phi (\boldsymbol{\theta}_t, \bar{\boldsymbol{z}}_t)) \triangleq \nabla_{\boldsymbol{\theta}^n} \Phi (\boldsymbol{\theta}_t, \bar{\boldsymbol{z}}_t) + \boldsymbol{e}_{\boldsymbol{\theta}^n, t} \quad (38)$$

and

$$\| \boldsymbol{e}_{\boldsymbol{\theta}^n, t} \|^2 \leq \| \nabla_{\boldsymbol{\theta}^n} \Phi (\boldsymbol{\theta}_t, \boldsymbol{z}_t^n) - \nabla_{\boldsymbol{\theta}^n} \Phi (\boldsymbol{\theta}_t, \bar{\boldsymbol{z}}_t) \|^2 \underset{(a)}{\leq} L^2 \| \boldsymbol{z}_t^n - \bar{\boldsymbol{z}}_t \|^2 \quad (39)$$

where (a) follows from Lemma 6, that implies that $\nabla \Phi (\boldsymbol{\theta}, \bar{\boldsymbol{z}})$ is Lipschitz continuous with parameter $L$ as a sum of two Lipschitz continuous functions. Let $\boldsymbol{e}_{\boldsymbol{\theta}, t} = \left[ \boldsymbol{e}_{\boldsymbol{\theta}^1, t}, \ldots, \boldsymbol{e}_{\boldsymbol{\theta}^N, t} \right]$ and $\boldsymbol{e}_t = [\boldsymbol{e}_{\boldsymbol{\theta}, t}, \boldsymbol{0}] \in \mathbb{R}^{\sum_{n=1}^{N} p_n + KQ}$. Let $\boldsymbol{g}_t = \left[ \boldsymbol{g}_t^1, \ldots, \boldsymbol{g}_t^N \right]$ and $\bar{\varphi}_t = \left[ \bar{\varphi}_t (x_1), \ldots, \bar{\varphi}_t (x_Q) \right]$ and recall that $\nabla_{\boldsymbol{\theta}} \Phi (\boldsymbol{\theta}_t, \bar{\boldsymbol{z}}_t) = \left[ \nabla_{\boldsymbol{\theta}^1} \Phi (\boldsymbol{\theta}_t, \bar{\boldsymbol{z}}_t), \ldots, \nabla_{\boldsymbol{\theta}^N} \Phi (\boldsymbol{\theta}_t, \bar{\boldsymbol{z}}_t) \right]$, $\nabla_{\bar{\boldsymbol{z}}} \Phi (\boldsymbol{\theta}_t, \bar{\boldsymbol{z}}_t) = \left[ \nabla_{\bar{\boldsymbol{z}}(x_1)} \Phi (\boldsymbol{\theta}_t, \bar{\boldsymbol{z}}_t), \ldots, \nabla_{\bar{\boldsymbol{z}}(x_Q)} \Phi (\boldsymbol{\theta}_t, \bar{\boldsymbol{z}}_t) \right]$ and $\nabla \Phi (\boldsymbol{\theta}_t, \bar{\boldsymbol{z}}_t) = \left[ \nabla_{\boldsymbol{\theta}} \Phi (\boldsymbol{\theta}_t, \bar{\boldsymbol{z}}_t), \nabla_{\bar{\boldsymbol{z}}} \Phi (\boldsymbol{\theta}_t, \bar{\boldsymbol{z}}_t) \right]$. Then, for all $t$, we have

$$\mathbb{E} \{ \Phi (\boldsymbol{\theta}_{t+1}, \bar{\boldsymbol{z}}_{t+1}) \mid \mathcal{F}_t \} \underset{(a)}{=} \mathbb{E} \{ \Phi (\boldsymbol{\theta}_t - \eta_t \boldsymbol{g}_t, \bar{\boldsymbol{z}}_t - \eta_t \bar{\varphi}_t) \mid \mathcal{F}_t \} \underset{(b)}{\leq}$$

$$\Phi (\boldsymbol{\theta}_t, \bar{\boldsymbol{z}}_t) - \eta_t (\nabla_{\boldsymbol{\theta}} \Phi (\boldsymbol{\theta}_t, \bar{\boldsymbol{z}}_t))^T \mathbb{E} \{ \boldsymbol{g}_t \mid \mathcal{F}_t \} - \eta_t (\nabla_{\bar{\boldsymbol{z}}} \Phi (\boldsymbol{\theta}_t, \bar{\boldsymbol{z}}_t))^T \mathbb{E} \{ \bar{\varphi}_t \mid \mathcal{F}_t \}$$

$$+ \frac{\eta_t^2}{2} L \mathbb{E} \left\{ \| \boldsymbol{g}_t \|^2 + \| \bar{\varphi}_t \|^2 \mid \mathcal{F}_t \right\} \underset{(c)}{\leq}$$

$$\Phi (\boldsymbol{\theta}_t, \bar{\boldsymbol{z}}_t) - \eta_t \| \nabla \Phi (\boldsymbol{\theta}_t, \bar{\boldsymbol{z}}_t) \|^2 - \eta_t (\nabla \Phi (\boldsymbol{\theta}_t, \bar{\boldsymbol{z}}_t))^T \boldsymbol{e}_t + \eta_t^2 L C_g \underset{(d)}{\leq}$$

$$\Phi (\boldsymbol{\theta}_t, \bar{\boldsymbol{z}}_t) - \eta_t \| \nabla \Phi (\boldsymbol{\theta}_t, \bar{\boldsymbol{z}}_t) \|^2 + \eta_t \| \nabla \Phi (\boldsymbol{\theta}_t, \bar{\boldsymbol{z}}_t) \| \| \boldsymbol{e}_t \| + \eta_t^2 L C_g \quad (40)$$

where (a) follows from (5) and (36), (b) uses the Lipschitz continuity of $\nabla \Phi (\boldsymbol{\theta}_t, \bar{\boldsymbol{z}}_t)$, so from Lemma 9

$$\Phi (\boldsymbol{y}) \leq \Phi (\boldsymbol{x}) + (\nabla \Phi (\boldsymbol{x}))^T (\boldsymbol{y} - \boldsymbol{x}) + \frac{1}{2} L \| \boldsymbol{y} - \boldsymbol{x} \|^2 \quad (41)$$

and also that $\Phi (\boldsymbol{\theta}_t, \bar{\boldsymbol{z}}_t)$ is $\mathcal{F}_t$-measurable. Inequality (c) uses (37) and (38) together with Lemma 5 with some $C_g > 0$, and inequality (d) is Cauchy-Schwarz.

Next, we want to apply the Robbins-Siegmund Theorem [44] on (40), with $z_t = \Phi (\boldsymbol{\theta}_t, \bar{\boldsymbol{z}}_t)$, $\beta_t = 0$, $\xi_t = \eta_t \| \nabla \Phi (\boldsymbol{\theta}_t, \bar{\boldsymbol{z}}_t) \| \| \boldsymbol{e}_t \| + \eta_t^2 L C_g$ and $\zeta_t = \eta_t \| \nabla \Phi (\boldsymbol{\theta}_t, \bar{\boldsymbol{z}}_t) \|^2$ (using the notation in [44]). To

that end, we show that for some constant $A > 0$

$$
\mathbb{E}\left\{\sum_{t=1}^{\infty} \eta_t \|\boldsymbol{e}_t\|\right\} \underset{(a)}{\leq} \sum_{t=1}^{\infty} \eta_t \mathbb{E}\left\{\|\boldsymbol{e}_t\|\right\} \leq \sum_{t=1}^{\infty} \eta_t \sqrt{\mathbb{E}\left\{\|\boldsymbol{e}_t\|^2\right\}} \underset{(b)}{\leq}
$$

$$
\sum_{t=1}^{\infty} \eta_t L \sqrt{\sum_{n=1}^{N} \mathbb{E}\left\{\|\boldsymbol{z}_t^n - \bar{\boldsymbol{z}}_t\|^2\right\}} \underset{(c)}{\leq} A \sum_{t=1}^{\infty} \eta_t \eta_{t-1} \underset{(d)}{<} \infty \quad (42)
$$

where (a) follows from Fatou's Lemma [45], (b) from (39) and (c) from Lemma 4. Inequality (d) follows from the limit comparison test [43] since we must have $\frac{\eta_t \eta_{t-1}}{\eta_t^2} = \frac{\eta_{t-1}}{\eta_t} \to 1$, and $\sum_{t=1}^{\infty} \eta_t^2 < \infty$. This follows since $\lim_{t \to \infty} \frac{\eta_{t+1}}{\eta_t}$ exists, and it must be equal to one otherwise the ratio test [43] will either say that $\sum_{t=1}^{\infty} \eta_t = \infty$ converges or that $\sum_{t=1}^{\infty} \eta_t^2 < \infty$ does not converge.

We conclude that $\sum_{t=1}^{\infty} \eta_t \|\boldsymbol{e}_t\| < \infty$ with probability 1. From (12) and (13) and the Lipschitz continuity of $\mathcal{L}_n(\boldsymbol{\theta}_t^n)$ and $s(\boldsymbol{\theta}_t^n, x)$ we conclude that $\|\nabla \Phi(\boldsymbol{\theta}_t, \bar{\boldsymbol{z}}_t)\|$ is bounded. Then, since $\sum_{t=1}^{\infty} \eta_t^2 < \infty$ we have $\sum_{t=1}^{\infty} \beta_t < \infty$ and $\sum_{t=1}^{\infty} \xi_t < \infty$ so [44] implies that with probability 1

$$
\sum_{t=1}^{\infty} \eta_t \|\nabla \Phi(\boldsymbol{\theta}_t, \bar{\boldsymbol{z}}_t)\|^2 < \infty. \quad (43)
$$

Now note that for some constant $D_g > 0$

$$
\|\nabla \Phi(\boldsymbol{\theta}_{t+1}, \bar{\boldsymbol{z}}_{t+1})\|^2 - \|\nabla \Phi(\boldsymbol{\theta}_t, \bar{\boldsymbol{z}}_t)\|^2 =
$$
$$
(\|\nabla \Phi(\boldsymbol{\theta}_{t+1}, \bar{\boldsymbol{z}}_{t+1})\| - \|\nabla \Phi(\boldsymbol{\theta}_t, \bar{\boldsymbol{z}}_t)\|)(\|\nabla \Phi(\boldsymbol{\theta}_{t+1}, \bar{\boldsymbol{z}}_{t+1})\| + \|\nabla \Phi(\boldsymbol{\theta}_t, \bar{\boldsymbol{z}}_t)\|) \leq
$$
$$
2L(\|\nabla \Phi(\boldsymbol{\theta}_{t+1}, \bar{\boldsymbol{z}}_{t+1})\| - \|\nabla \Phi(\boldsymbol{\theta}_t, \bar{\boldsymbol{z}}_t)\|) \leq 2L(\|\nabla \Phi(\boldsymbol{\theta}_{t+1}, \bar{\boldsymbol{z}}_{t+1}) - \nabla \Phi(\boldsymbol{\theta}_t, \bar{\boldsymbol{z}}_t)\|) \leq
$$
$$
2L^2 \|(\boldsymbol{\theta}_{t+1}, \bar{\boldsymbol{z}}_{t+1}) - (\boldsymbol{\theta}_t, \bar{\boldsymbol{z}}_t)\| = 2L^2 \|(\eta_t g_t, \eta_t \bar{\boldsymbol{\varphi}}_t)\| \leq D_g \eta_t. \quad (44)
$$

Combining (43) and (44), [46, Proposition 2] with $\alpha_k = \eta_k$ and $\beta_k = \|\nabla \Phi(\boldsymbol{\theta}_k, \bar{\boldsymbol{z}}_k)\|^2$ confirms that $\|\nabla \Phi(\boldsymbol{\theta}_k, \bar{\boldsymbol{z}}_k)\| \to 0$ with probability 1, so $\|\nabla_{\bar{\boldsymbol{z}}(x)} \Phi(\boldsymbol{\theta}_t, \bar{\boldsymbol{z}}_t)\| \to 0$ and $\|\nabla_{\boldsymbol{\theta}^n} \Phi(\boldsymbol{\theta}_t, \bar{\boldsymbol{z}}_t)\| \to 0$ with probability 1 as well, for each $x \in \mathcal{D}_r$ and $n$. Therefore, for each $x \in \mathcal{D}_r$ and each $n$

$$
\|\nabla_{\bar{\boldsymbol{z}}(x)} \Phi(\boldsymbol{\theta}_t, \bar{\boldsymbol{z}}_t)\| = 2\beta \left\| \bar{\boldsymbol{z}}_t(x) - \frac{1}{N} \sum_{n=1}^{N} s(\boldsymbol{\theta}_t^n, x) \right\| \to 0. \quad (45)
$$

Let $\bar{s}(\boldsymbol{\theta}_t, x) \triangleq \frac{1}{N} \sum_{n=1}^{N} s(\boldsymbol{\theta}_t^n, x)$. Hence, we conclude that for each $n$

$$
\left\| \nabla_{\boldsymbol{\theta}^n} \left( \mathcal{L}_n(\boldsymbol{\theta}_t^n) + \frac{\beta}{N-1} \sum_{x \in \mathcal{D}_r} \|\bar{s}(\boldsymbol{\theta}_t, x) - s(\boldsymbol{\theta}_t^n, x)\|^2 \right) \right\| =
$$

$$
\left\| \nabla \mathcal{L}_n(\boldsymbol{\theta}_t^n) + \frac{2\beta}{N} \sum_{x \in \mathcal{D}_r} (\nabla s(\boldsymbol{\theta}_t^n, x))^T (s(\boldsymbol{\theta}_t^n, x) - \bar{s}(\boldsymbol{\theta}_t, x)) \right\| \leq
$$

$$
\left\| \frac{2\beta}{N} \sum_{x \in \mathcal{D}_r} (\nabla s(\boldsymbol{\theta}_t^n, x))^T ((s(\boldsymbol{\theta}_t^n, x) - \bar{s}(\boldsymbol{\theta}_t, x)) - (s(\boldsymbol{\theta}_t^n, x) - \bar{\boldsymbol{z}}_t(x))) \right\| +
$$

$$
\left\| \nabla \mathcal{L}_n(\boldsymbol{\theta}_t^n) + \frac{2\beta}{N} \sum_{x \in \mathcal{D}_r} (\nabla s(\boldsymbol{\theta}_t^n, x))^T (s(\boldsymbol{\theta}_t^n, x) - \bar{\boldsymbol{z}}_t(x)) \right\| \underset{(a)}{\leq}
$$

$$
\frac{2\beta}{N} \sum_{x \in \mathcal{D}_r} L_s \|\bar{\boldsymbol{z}}_t(x) - \bar{s}(\boldsymbol{\theta}_t, x)\| + \|\nabla_{\boldsymbol{\theta}^n} \Phi(\boldsymbol{\theta}_t, \bar{\boldsymbol{z}}_t)\| \to 0 \quad (46)
$$

where (a) follows from the triangle inequality and the $L_s$−Lipschitz continuity of $s(\boldsymbol{\theta}_t^n, x)$. $\qquad \square$

## 11 Auxiliary Lemmas

The next lemma characterizes the spectral properties of the disagreement matrix, used in Lemma 4.

**Lemma 7.** *Let $\tilde{W} = W \otimes I_K$, where $I_K$ is the identity matrix of size $K$. Let $\lambda_i(A)$ be the $i$-th largest eigenvalue of $A$. Then for every $\boldsymbol{v} = \left[\boldsymbol{v}^1, ..., \boldsymbol{v}^N\right] \in \mathbb{R}^{KN}$ such that $\boldsymbol{v}^n \in \mathbb{R}^K$ for each $n$:*

1. $\left\|\left(I - \tilde{W}\right)\boldsymbol{v}\right\|^2 \leq 4\left\|\boldsymbol{v}\right\|^2.$

2. $\left\|\left(I - \tilde{W}\right)\boldsymbol{v}\right\|^2 \geq \lambda_2\left((I-W)^T(I-W)\right)\left\|\boldsymbol{v} - \mathbf{1}_N \otimes \frac{1}{N}\sum_{m=1}^N \boldsymbol{v}^m\right\|^2$ *and* $\lambda_2\left((I-W)^T(I-W)\right) > 0.$

3. $\left\|\tilde{W}\left(I - \tilde{W}\right)\boldsymbol{v}\right\|^2 \leq \lambda_{N-1}\left(W^T W\right)\left\|\left(I - \tilde{W}\right)\boldsymbol{v}\right\|^2$ *and* $\lambda_{N-1}\left(W^T W\right) < 1.$

*Proof.* The sum of each row in the non-negative matrix $W + W^T$ is 2 since $W$ and $W^T$ are stochastic. Furthermore, $W + W^T$ is an irreducible matrix since $G$ is strongly connected. Hence, the Perron-Frobenius Theorem [47, Theorem 8.4.4, Page 534] yields $\lambda_{\min}\left(W^T + W\right) \geq -2$. Since $W^T W \mathbf{1} = W^T \mathbf{1} = \mathbf{1}$, $W^T W$ is also a stochastic matrix. Let $\tilde{w}_{n,m}$ be an element of $W^T W$, then

$$\tilde{w}_{n,m} = \sum_{i=1}^N w_{n,i} w_{m,i} \geq w_{n,m} w_{m,m} \tag{47}$$

so if $w_{n,n} > 0$ for all $n$, then $\tilde{w}_{n,m} > 0$ if $w_{n,m} > 0$. Hence, $W^T W$ is also irreducible and the Perron-Frobenius Theorem yields $\lambda_{\max}\left(W^T W\right) = 1$. Then we get from the Rayleigh quotient that

$$\lambda_{\max}\left((I-W)^T(I-W)\right) = \max_{\boldsymbol{v} \neq 0} \frac{\boldsymbol{v}^T (I-W)^T (I-W)\boldsymbol{v}}{\boldsymbol{v}^T \boldsymbol{v}} \leq$$

$$1 - \min_{\boldsymbol{v} \neq 0} \frac{\boldsymbol{v}^T \left(W + W^T\right)\boldsymbol{v}}{\boldsymbol{v}^T \boldsymbol{v}} + \max_{\boldsymbol{v} \neq 0} \frac{\boldsymbol{v}^T W^T W \boldsymbol{v}}{\boldsymbol{v}^T \boldsymbol{v}} \leq 4 \tag{48}$$

which shows that $\lambda_{\max}\left(\left(I_{KN} - \tilde{W}\right)^T \left(I_{KN} - \tilde{W}\right)\right) \leq 4$ since the singular values of $I_{KN} - \tilde{W}$ are that of $I - W$, each with multiplicity $K$. This proves part 1.

Since $W$ is a stochastic matrix and $G$ is strongly connected, then the Perron-Frobenius Theorem states that $\lambda_{\max}(W) = 1$ with eigenspace $\{\alpha\mathbf{1} \mid \alpha \in \mathbb{R}\}$. Hence $\lambda_{\min}(I - W) = 0$ with eigenspace $\{\alpha\mathbf{1} \mid \alpha \in \mathbb{R}\}$. Then the null-space of $I_{KN} - \tilde{W}$ has dimension $K$, and we can check it is $\text{span}\{\mathbf{1}_N \otimes \boldsymbol{e}_i \mid 1 \leq i \leq K\}$ where $\boldsymbol{e}_i$ is a standard basis vector of $\mathbb{R}^K$, simply because $\tilde{W}\left(\mathbf{1}_N \otimes \boldsymbol{e}_i\right) = \mathbf{1}_N \otimes \boldsymbol{e}_i$ for each $1 \leq i \leq K$. Let $\boldsymbol{u}_1, ..., \boldsymbol{u}_{KN}$ be the orthogonal sorted eigenvectors of the symmetric $\left(I_{KN} - \tilde{W}\right)^T \left(I_{KN} - \tilde{W}\right)$. Note that if $\left(I_{KN} - \tilde{W}\right)^T \left(I_{KN} - \tilde{W}\right)\boldsymbol{v} = \mathbf{0}$ then $\left\|\left(I_{KN} - \tilde{W}\right)\boldsymbol{v}\right\| = 0$ so $\left(I_{KN} - \tilde{W}\right)\boldsymbol{v} = \mathbf{0}$. Hence the null-space of $\left(I_{KN} - \tilde{W}\right)^T \left(I_{KN} - \tilde{W}\right)$, spanned by $\boldsymbol{u}_1, ..., \boldsymbol{u}_K$, is that of $I_{KN} - \tilde{W}$, which is $\text{span}\{\mathbf{1}_N \otimes \boldsymbol{e}_i \mid 1 \leq i \leq K\}$. Since $\left\langle \mathbf{1}_N \otimes \boldsymbol{e}_i, \boldsymbol{v} - \mathbf{1}_N \otimes \frac{1}{N}\sum_{m=1}^N \boldsymbol{v}^m \right\rangle = 0$ for every $1 \leq i \leq K$ (sum of $i$-th coordinates is zero), then $\boldsymbol{v} - \mathbf{1}_N \otimes \frac{1}{N}\sum_{m=1}^N \boldsymbol{v}^m \in \text{span}\{\boldsymbol{u}_{K+1}, ..., \boldsymbol{u}_{KN}\}$, which gives part 2 since $\left(I_{KN} - \tilde{W}\right)\boldsymbol{v} = \left(I_{KN} - \tilde{W}\right)\left(\boldsymbol{v} - \mathbf{1}_N \otimes \frac{1}{N}\sum_{m=1}^N \boldsymbol{v}^m\right)$, which follows from $((I-W) \otimes I_K)(\mathbf{1}_N \otimes \boldsymbol{v}) = (I-W)\mathbf{1}_N \otimes \boldsymbol{v} = 0$ (the mixed-product property and $(I-W)\mathbf{1}_N = 0$).

Now let $\boldsymbol{b}_1, ..., \boldsymbol{b}_{KN}$ be the orthogonal sorted eigenvectors of the symmetric $\tilde{W}^T\tilde{W}$. From Perron-Frobenius Theorem, the eigenspace of $\lambda_{\max}\left(W^T W\right) = 1$ is $\{\alpha\mathbf{1} \mid \alpha \in \mathbb{R}\}$. Hence $\tilde{W} = W \otimes I_K$ has the singular value $\lambda_{\max}\left(W^T W\right) = 1$ with multiplicity $K$. Again we can check that the eigenspace of $\lambda_{\max}\left(W^T W\right) = 1$ is $\text{span}\{\mathbf{1}_N \otimes \boldsymbol{e}_i \mid 1 \leq i \leq K\}$, simply because $\tilde{W}^T\tilde{W}\left(\mathbf{1}_N \otimes \boldsymbol{e}_i\right) = \mathbf{1}_N \otimes \boldsymbol{e}_i$ for each $1 \leq i \leq K$. Since $\left\langle \mathbf{1}_N \otimes \boldsymbol{e}_i, \left(I_{KN} - \tilde{W}\right)\boldsymbol{v} \right\rangle = \boldsymbol{v}^T\left(I_{KN} - \tilde{W}^T\right)\left(\mathbf{1}_N \otimes \boldsymbol{e}_i\right) = 0$ for every $1 \leq i \leq K$, we have $\left(I_{KN} - \tilde{W}\right)\boldsymbol{v} \in \text{span}\{\boldsymbol{b}_1, ..., \boldsymbol{b}_{K(N-1)}\}$, which gives part 3. $\square$

The next simple lemma is used in many of our proofs.

**Lemma 8.** *For every $n > 0$ we have $\left\| \sum_{i=1}^{n} \boldsymbol{y}_i \right\|^2 \leq n \sum_{i=1}^{n} \|\boldsymbol{y}_i\|^2$.*

*Proof.* We prove this result by induction on $n$. For $n = 1$ it is trivial. Now assume that the inequality holds for all $l \leq n - 1$. Then

$$\left\| \sum_{i=1}^{n} \boldsymbol{y}_i \right\|^2 = \left\| \sum_{i=1}^{\lfloor \frac{n}{2} \rfloor} \boldsymbol{y}_i + \sum_{i=\lfloor \frac{n}{2} \rfloor + 1}^{n} \boldsymbol{y}_i \right\|^2 \underset{(a)}{\leq}$$

$$\left\| \sum_{i=1}^{\lfloor \frac{n}{2} \rfloor} \boldsymbol{y}_i \right\|^2 + 2 \left\| \sum_{i=1}^{\lfloor \frac{n}{2} \rfloor} \boldsymbol{y}_i \right\| \left\| \sum_{i=\lfloor \frac{n}{2} \rfloor + 1}^{n} \boldsymbol{y}_i \right\| + \left\| \sum_{i=\lfloor \frac{n}{2} \rfloor + 1}^{n} \boldsymbol{y}_i \right\|^2 \underset{(b)}{\leq} 2 \left\| \sum_{i=1}^{\lfloor \frac{n}{2} \rfloor} \boldsymbol{y}_i \right\|^2 + 2 \left\| \sum_{i=\lfloor \frac{n}{2} \rfloor + 1}^{n} \boldsymbol{y}_i \right\|^2 \underset{(c)}{\leq}$$

$$2 \left\lfloor \frac{n}{2} \right\rfloor \sum_{i=1}^{\lfloor \frac{n}{2} \rfloor} \|\boldsymbol{y}_i\|^2 + 2 \left( n - \left\lfloor \frac{n}{2} \right\rfloor \right) \sum_{i=\lfloor \frac{n}{2} \rfloor + 1}^{n} \|\boldsymbol{y}_i\|^2 = n \sum_{i=1}^{\lfloor \frac{n}{2} \rfloor} \|\boldsymbol{y}_i\|^2 + n \sum_{i=\lfloor \frac{n}{2} \rfloor + 1}^{n} \|\boldsymbol{y}_i\|^2$$

$$+ \left( n - 2 \left\lfloor \frac{n}{2} \right\rfloor \right) \left( \sum_{i=\lfloor \frac{n}{2} \rfloor + 1}^{n} \|\boldsymbol{y}_i\|^2 - \sum_{i=1}^{\lfloor \frac{n}{2} \rfloor} \|\boldsymbol{y}_i\|^2 \right) \underset{(d)}{\leq} n \sum_{i=1}^{n} \|\boldsymbol{y}_i\|^2 \quad (49)$$

where (a) follows from Cauchy-Schwarz, (b) since $(x - y)^2 \geq 0$ and (c) from the induction hypothesis. Inequality (d) assumes that without loss of generality, $\sum_{i=1}^{\lfloor \frac{n}{2} \rfloor} \|\boldsymbol{y}_i\|^2 \geq \sum_{i=\lfloor \frac{n}{2} \rfloor + 1}^{n} \|\boldsymbol{y}_i\|^2$. $\square$

The next Lemma is a well known bound for functions with Lipschitz gradients. We provide the proof here for completeness.

**Lemma 9.** *If $\Phi : \mathbb{R}^d \to \mathbb{R}$ has Lipschitz continuous gradients with parameter L, then for every $\boldsymbol{x}, \boldsymbol{y} \in \mathbb{R}^d$*

$$\Phi(\boldsymbol{y}) \leq \Phi(\boldsymbol{x}) + (\nabla \Phi(\boldsymbol{x}))^T (\boldsymbol{y} - \boldsymbol{x}) + \frac{1}{2} L \|\boldsymbol{y} - \boldsymbol{x}\|^2. \quad (50)$$

*Proof.* By Cauchy-Schwartz

$$(\nabla \Phi(\boldsymbol{y}) - \nabla \Phi(\boldsymbol{x}))^T (\boldsymbol{y} - \boldsymbol{x}) \leq \|\nabla \Phi(\boldsymbol{y}) - \nabla \Phi(\boldsymbol{x})\| \|\boldsymbol{y} - \boldsymbol{x}\| \leq L \|\boldsymbol{y} - \boldsymbol{x}\|^2. \quad (51)$$

Now define $g(\boldsymbol{x}) = \frac{L}{2} \|\boldsymbol{x}\|^2 - \Phi(\boldsymbol{x})$, then

$$(\nabla g(\boldsymbol{y}) - \nabla g(\boldsymbol{x}))^T (\boldsymbol{y} - \boldsymbol{x}) = L \|\boldsymbol{y} - \boldsymbol{x}\|^2 - (\nabla \Phi(\boldsymbol{y}) - \nabla \Phi(\boldsymbol{x}))^T (\boldsymbol{y} - \boldsymbol{x}) \geq 0 \quad (52)$$

so $g(\boldsymbol{x})$ is convex, as a differentiable function. Then

$$g(\boldsymbol{y}) \geq g(\boldsymbol{x}) + (\nabla g(\boldsymbol{x}))^T (\boldsymbol{y} - \boldsymbol{x}) \quad (53)$$

which is just

$$\frac{L}{2} \|\boldsymbol{y}\|^2 - \Phi(\boldsymbol{y}) \geq \frac{L}{2} \|\boldsymbol{x}\|^2 - \Phi(\boldsymbol{x}) + (L\boldsymbol{x} - \nabla \Phi(\boldsymbol{x}))^T (\boldsymbol{y} - \boldsymbol{x}) \implies$$

$$\frac{L}{2} \|\boldsymbol{y}\|^2 + \frac{L}{2} \|\boldsymbol{x}\|^2 - L\boldsymbol{x}^T \boldsymbol{y} + \Phi(\boldsymbol{x}) + (\nabla \Phi(\boldsymbol{x}))^T (\boldsymbol{y} - \boldsymbol{x}) \geq \Phi(\boldsymbol{y}) \implies$$

$$\Phi(\boldsymbol{y}) \leq \Phi(\boldsymbol{x}) + (\nabla \Phi(\boldsymbol{x}))^T (\boldsymbol{y} - \boldsymbol{x}) + \frac{1}{2} L \|\boldsymbol{y} - \boldsymbol{x}\|^2. \quad (54)$$

$\square$

The next lemma shows that given our assumptions on the differentiability of the loss function and the smoothness of the soft-decision function of the classifier, the private loss function of each device is Lipschitz continuous with Lipschitz continuous gradients. The importance is merely technical, and is meant to compress our set of assumption. Instead, one can just assume that the private loss function is Lipschitz continuous with Lipschitz continuous gradients. As our simulations suggest, our algorithm converges with practical DNNs regardless of their smoothness properties.

**Lemma 10.** *Assume that $\mathcal{L}(s, \boldsymbol{y})$ is twice continuously differentiable on $\Delta^K$ (with respect to $s$) and that $s(\boldsymbol{\theta}^n, x)$ is Lipschitz continuous with Lipschitz continuous gradients. Then $\mathcal{L}_n(\boldsymbol{\theta}^n) = \sum_{\tilde{x} \in \mathcal{D}_n} \mathcal{L}(s(\boldsymbol{\theta}^n, \tilde{x}), \boldsymbol{y}(\tilde{x}))$ is Lipschitz continuous with Lipschitz continuous gradients.*

*Proof.* If $\mathcal{L}(s, \boldsymbol{y}) : \Delta^K \times \Delta^K \to \mathbb{R}_0^+$ is twice continuously differentiable in $s$ (as vector function) then since the simplex $\Delta^K$ is closed, then by the extreme value theorem (see [48, Page 89]) $\|\nabla_s \mathcal{L}_n(s, \boldsymbol{y})\| \leq M_1$ for some $M_1 > 0$ and the Hessian satisfies $\|\nabla_s^2 \mathcal{L}_n(s, \boldsymbol{y})\| \leq M_2$ for some $M_2 > 0$. Then by the mean value theorem (see [48, Page 127]) $\mathcal{L}(s, \boldsymbol{y})$ is Lipschitz continuous with Lipschitz continuous gradients, for every $\boldsymbol{y} \in \Delta^K$.

Then since $s(\boldsymbol{\theta}^n, x)$ is Lipschitz continuous, the composition $\mathcal{L}_n(\boldsymbol{\theta}^n)$ is also Lipschitz continuous. Additionally, $\mathcal{L}_n(\boldsymbol{\theta}^n)$ has Lipschitz continuous gradients since for any $\boldsymbol{\theta}_1, \boldsymbol{\theta}_2 \in \mathbb{R}^{p_n}$

$$\|\nabla \mathcal{L}_n(\boldsymbol{\theta}_2) - \nabla \mathcal{L}_n(\boldsymbol{\theta}_1)\| \leq$$

$$\left\| \sum_{\tilde{x} \in \mathcal{D}_n} (\nabla s(\boldsymbol{\theta}_1, x))^T \nabla_s \mathcal{L}(s(\boldsymbol{\theta}_1, \tilde{x}), \boldsymbol{y}(\tilde{x})) - \sum_{\tilde{x} \in \mathcal{D}_n} (\nabla s(\boldsymbol{\theta}_2, x))^T \nabla_s \mathcal{L}(s(\boldsymbol{\theta}_2, \tilde{x}), \boldsymbol{y}(\tilde{x})) \right\| \leq$$

$$\left\| \sum_{\tilde{x} \in \mathcal{D}_n} (\nabla s(\boldsymbol{\theta}_1, x) - \nabla s(\boldsymbol{\theta}_2, x))^T \nabla_s \mathcal{L}(s(\boldsymbol{\theta}_1, \tilde{x}), \boldsymbol{y}(\tilde{x})) \right\| +$$

$$\left\| \sum_{\tilde{x} \in \mathcal{D}_n} (\nabla s(\boldsymbol{\theta}_2, x))^T (\nabla_s \mathcal{L}(s(\boldsymbol{\theta}_1, \tilde{x}), \boldsymbol{y}(\tilde{x})) - \nabla_s \mathcal{L}(s(\boldsymbol{\theta}_2, \tilde{x}), \boldsymbol{y}(\tilde{x}))) \right\| \underset{(a)}{\leq}$$

$$\sum_{\tilde{x} \in \mathcal{D}_n} \|\nabla s(\boldsymbol{\theta}_1, x) - \nabla s(\boldsymbol{\theta}_2, x)\| \|\nabla_s \mathcal{L}(s(\boldsymbol{\theta}_1, \tilde{x}), \boldsymbol{y}(\tilde{x}))\| +$$

$$\sum_{\tilde{x} \in \mathcal{D}_n} \|\nabla s(\boldsymbol{\theta}_2, x)\| \|\nabla_s \mathcal{L}(s(\boldsymbol{\theta}_1, \tilde{x}), \boldsymbol{y}(\tilde{x})) - \nabla_s \mathcal{L}(s(\boldsymbol{\theta}_2, \tilde{x}), \boldsymbol{y}(\tilde{x}))\| \underset{(b)}{\leq} L \|\boldsymbol{\theta}_2 - \boldsymbol{\theta}_1\| \quad (55)$$

where (a) uses the triangle inequality and the definition of the spectral norm for the matrix $\nabla s(\boldsymbol{\theta}, x)$. Inequality (b) uses that $s(\boldsymbol{\theta}_2, x)$ and $\mathcal{L}(s, \boldsymbol{y})$ (with respect to $s$) are both Lipschitz continuous with Lipschitz continuous gradients. □

## 12 Simulations

### 12.1 Simulation Setup

Across the simulations four models were used: `LeNet-5` [40], `ResNet-2`, `ResNet-8`, and `ResNet-14` [42]. The `ResNet-14` model is built from the original `ResNet-18` architecture in [42] but with the lower number of filters {16, 32, 64} and three residual blocks used for `ResNet-20` on `CIFAR10` in [42]. The `ResNet-2` and `ResNet-8` were then similarly constructed, but with only one and two residual blocks respectively.

Simulations were run on both the `MNIST` dataset [40] and the `CIFAR10` dataset. For `MNIST` we used the 60k/10k training/test split as the dataset is packaged. We used the 50k/10k training/test split of `CIFAR10`. For both datasets we further divided the training set as detailed below for the different simulation settings.

Keeping in line with the analysis we used plain SGD without momentum or additional regularization methods for all algorithms, which maintains a fair comparison. Cross-entropy was used as the private loss function for all simulations.

Each doubly stochastic communication matrix was randomly generated given a maximum degree parameter and constructed from a random convex combination of random permutation matrices, based on Birkhoff's Theorem (pg. 549) [47].

For the setting of devices training `LeNet-5` on `MNIST` which includes Fig. 2 and Fig. 5, first a random 40% of the training data points were withheld and used as the reference dataset that all devices receive. The remaining 60% was divided uniformly at random, without replacement over the 16 devices as their private training data. In the simulations where there were less than 16 devices we maintained

the same division of data points. This means that as we went from 2 to 4 devices and 4 to 8 devices, etc. we were simply increasing the number of devices in the communication network. Relatedly, in both Fig. 2 and Fig. 5 all communication networks had a maximum degree of 3, except for when there were only 2 devices which had a maximum degree of 1. The private data batch sizes for SGD with all algorithms was 32 and the network batch size for D-Distillation was likewise 32.

Hyper-parameters tuning was conducted for a communication network of 8 devices, and then the selected hyper-parameters were used for the other networks with different numbers of devices in this setting. The selection was done to maximize the final validation accuracy from $80\%/20\%$ training/validation split of each device's private data after 800 epochs. The entire device's private training data was then used for the final training round where the test results were evaluated. The private and network batch sizes were set to 128 in this setting unless labeled otherwise. For each algorithm (D-Distillation, D-SGD, Silo-SGD) we tested starting step sizes of $\{0.01, 0.02, 0.05, 0.1, 0.15, 0.2, 0.5\}$ and values of the regularization parameter $\beta$ for D-Distillation of $\{1, 3, 6, 12\}$. This led to the selection of a starting step size of $0.15$ for all three algorithms and $\beta = 3$ for D-Distillation. All simulations were run with a step size schedule that decreased by an order of magnitude after 150 and 225 epochs. After 225 epochs a decay multiple, $e^{-0.55(m-400)}$ for epoch number $m$, was included to abide by the learning rate / step size assumptions of Theorem 3.

The MNIST results in Fig. 4 used the same settings as above. There was no further optimization for any of the communication reduction schemes.

The training of 8 devices with `ResNet-14` on `CIFAR10` for Fig. 4 followed the same training outline as above. We trained both D-SGD and D-Distillation without any of the communication reduction schemes and on a communication network of 8 devices with a maximum degree of 4. A random $25\%$ of the training data points were withheld and used as the reference dataset, and the rest were randomly assigned to the 8 devices as private data. Once the hyper-parameters had been selected for D-Distillation, the same hyper-parameters were used for each of communication reduction schemes without further optimization. Optimizing for mean validation accuracy at 800 epochs, we tested starting step sizes of $\{0.01, 0.02, 0.05, 0.1\}$ and values of the regularization parameter $\beta$ for D-Distillation of $\{1, 6, 12, 16, 24\}$. This led to the selection of a starting step size of $0.02$ for all three algorithms and $\beta = 16$ for D-Distillation. All simulations were run with a step size schedule that decreased by an order of magnitude after 300 and 600 epochs. After 600 epochs a decay multiple, $e^{-0.55(m-600)}$ for epoch number $m$, was included.

For the cross-model architecture simulations of Fig. 3 on `CIFAR10` we followed the same training outline as above, but we tuned the hyper-parameters separately for the `LeNet-5` and `ResNet-8` on 8 devices. `ResNet-2` was run with the selected hyper-parameters for `ResNet-8` with no further tuning. The optimization of `ResNet-14` occurred in the previous setting and the same parameters were used here. `LeNet-5` was run over the starting step sizes of $\{0.01, 0.02, 0.05, 0.1, 0.15, 0.2, 0.5\}$ and $\beta$ for D-Distillation of $\{1, 3, 6, 12, 16, 24\}$. This led to the selection of a starting step size of $0.15$ for D-SGD and $0.05$ for Silo-SGD and D-Distillation, with $\beta = 6$. `ResNet-8` was run over the starting step sizes of $\{0.02, 0.05, 0.1, 0.15, 0.2\}$ and $\beta$ for D-Distillation of $\{1, 3, 6, 12, 24\}$. Ultimately, $0.15$ was selected as the initial step size for D-SGD and Silo-SGD, while $0.05$ with $\beta = 6$ was selected for D-Distillation. All simulations were run with a step size schedule that decreased by an order of magnitude after 300 and 600 epochs. After 600 epochs a decay multiple, $e^{-0.55(m-600)}$ for epoch number $m$, was included.

For the communication network in the cross-model simulations, D-Distillation had one 8 device network with a maximum degree of 3 with devices of different model types assigned at random to their position in the network. Since D-SGD cannot run between different models, we instead randomly assigned devices to one of two or one of four model types. Then we formed one communication network per model architecture for D-SGD. When there were two model types, and therefore 4 devices in each of the two D-SGD communication networks they were assigned a maximum degree of 3. When there were four model types, and therefore 2 devices in each of the two D-SGD communication networks they were assigned a maximum degree of 1.

The final setting pertains to Fig. 6 and Fig. 7 where `ResNet-14` was training on `CIFAR10`. A smaller number of private data points (750 per device) were allocated, and instead increasing sizes of reference data sets were simulated. We note that in practice the size of the reference data set does not come at a decrease to the private data set, especially since the reference data set is unlabeled and can come from a different source. In this setting, 6 devices were simulated on a communication graph with a

maximum degree of 3. Both private and network batch sizes were set to 32 for all runs. We performed hyper-parameter selection on 2 devices and for simplicity maintained the same setting with 6 devices. The selection process used a $80\%/20\%$ training/validation split and then combined the two subsets for the final training and test evaluation round. We selected the initial step size from the same list as above after 800 total epochs and with $\beta$ values across $\{1, 2, 3, 6, 12, 24\}$. This led to step size selection of 0.02 for all algorithms with a $\beta = 6$ for D-Distillation. All simulations were run with a step size schedule that decreased by an order of magnitude after 400 and 600 epochs, after also testing decreases at 200/300 and 300/450. After 600 epochs a decay multiple, $e^{-0.55(m-600)}$ for epoch number $m$, was added to abide by the step size assumptions of Theorem 3. Reference data set sizes of 750, 1500 and 7500 were evaluated, and 7500 was selected for the results displayed in Fig. 7.

The simulations were run on a server with 40 Intel Xeon CPU E5-2630 v4 at 2.20GHz and 252GB system memory. Simulations with `LeNet-5` on 2 devices took 0.5 hours, 4 devices took 1 hour, 8 devices took 6, and 16 devices took 30 hours. Simulations with `ResNet-14` on 2 devices took 1.2 hours, 6 devices took 6 hours, 8 devices took 12 hours.

## 12.2 Additional Simulation Results

### 12.2.1 Varying Number of Devices

The setting of edge devices is different from traditional centralized training or even distributed methods for data parallelism. The real-world setting of edge devices with private data means that the amount of data per device is a given. The only options are to let each device train on its own (Silo-SGD) or see if collaboration (without private data sharing) can improve each device's training. The more devices that can be included in a communication network to participate in training with an effective collaborative algorithm, the more private data the network will have from which to distill knowledge. Simulations below compare the performance for 2, 4, 8 and 16 devices in the communication network. The results of Fig. 5 display a general accuracy improvement from adding more devices to the network. These initial findings indicate that the D-Distillation algorithm will likely continue to increase its performance gains over Silo-SGD for larger communication networks. Additional simulations with more devices will provide further insight into how the performance of D-Distillation scales with the number of devices.

Figure 5: Test accuracy mean across devices for 2, 4, 8 and 16 devices in the communication network with the same number of private data points per device, network batch sizes of 32 for the `LeNet-5` models training on `MNIST`.

### 12.2.2 Varying Reference Data Set Size

Even when the private data set size is fixed, the algorithm designer can still curate both the data points and adjust the size of the reference data set for D-Distillation. The simulations in Fig. 6 evaluate the accuracy and each loss term involved in D-Distillation for three reference data set sizes of $\{750, 1500, 7500\}$ data points while training `ResNet-14` on `CIFAR10`. A smaller number of private data points (750 per device) are allocated to allow for the increasing sizes of reference data sets from the `CIFAR10` dataset. In practice, an increased reference data set size does not require a decrease in the private data set, especially since the reference data set is unlabeled and can come from a different source. All other hyper-parameters are consistent over this set of simulations. Since each of the

6 devices in these simulations has 750 data points it naturally leads to a much lower baseline test accuracy than the previous simulations on `CIFAR10` in Fig. 3 and Fig. 4. The main point of this experiment is to quantify the relative gain of increasing the reference data set size, which comes at no additional computation or communication cost.

Encouragingly, we find that the test accuracy sees a *6.85 percentage point improvement* for a $10\times$ larger reference data set. Interestingly, the average private loss (cross-entropy in (1)) on the test data experiences a more drastic reduction of 51.5%. To understand the test private loss results, we examine the loss terms from the training process. The soft-decision loss (the second term in (2) between the reference soft-decision and the network soft-decision) is consistently highest for the largest reference dataset. The network loss is analogous to the soft-decision loss, but it is the mean-squared error between the local device's network soft-decisions and the weighted average of its neighbors' communicated network soft-decisions. In the simulation results, the network loss for the largest reference data set is particularly differentiated in its sustained high value until the step size decreases rapidly after 400 and 600 epochs. When these rapid drops appear in both the network and soft-decisions losses, they also appear in the test private loss. This indicates the sustained disagreement between reference soft-decisions and network soft-decisions across devices in the first half of the training process is related to the improved performance with a larger reference dataset (and the greater variety of data points included within). A larger reference data set is just one mechanism to achieve this variety and sustained disagreement during the early training regime.

Figure 6: D-Distillation accuracy and different loss term values for varying reference dataset size. The simulations shown are for 6 devices with `ResNet-14` models on a communication graph with a maximum degree of 3 and a per device private dataset size of 750 data points on `CIFAR10`.

### 12.2.3 Hyper-Parameter Grid-Search for Two Devices

Finally, we exploited the short run time of the small private data set size for 2 devices to perform a full grid-search over the step size, step size decreases, beta, and reference data set size for `ResNet-14` on `CIFAR10`. The best results are presented in Fig. 7 where D-Distillation reaches the same final accuracy as D-SGD but with a $160\times$ reduction in communication. In this setting, the performance of D-Distillation coincides with that of D-SGD and suggests that further D-Distillation performance gains can be achieved from refined hyper-parameter tuning in other data and device regimes.

Figure 7: Average test accuracy by epoch (left) and total communication (right) for Silo-SGD, D-SGD, D-Distillation for a 2 device communication network and 750 data points per device after grid-search based tuning.

## 12.3 Possible Performance Enhancements

In this subsection we discuss the key ideas to explore in order to unleash the full practical potential of D-distillation, based on what theory and our initial simulation results suggest.

Possible performance enhancements include the following:

*Number of devices*: The motivation of distributed on-device learning is to efficiently distill knowledge from across the network of devices. We expect that the more devices in the network (i.e., more private data in the network), the greater the improvement will be compared to Silo-SGD. As seen in Fig. 5, our simulations agree with this theoretical expectation; the performance increases with the number of devices. Furthermore, D-Distillation's cross-architecture learning ability, which D-SGD (or FL) does not share, allows more devices to join the distributed training process and thereby improve performance.

*Reference data set*: Our reference data set is unlabeled and can consist of artificiality designed data points. Ensuring the reference dataset has a balanced number of data points over the different classes is a basic requirement. However, introducing smart structure to these data points (e.g., spanning the space of likely features) has the potential to improve performance significantly. This requires a separate investigation that may benefit from ideas in coding theory.

*Reference data set size*: As can be seen in Fig. 6, the reference data set size has a significant impact on performance. Our theoretical analysis suggests that a larger reference dataset allows for devices to learn more from each other as the set of stationary points of (3) becomes more selective. However, focusing too much on agreeing with other devices during the training process may sacrifice generalization accuracy for the sole purpose of agreeing on arbitrary consensus values. This tradeoff is controlled by the $\beta$ *parameter* of equation (5). Optimizing the balance between learning from private data and the network is a straightforward way to improve performance.

*Network batch size*: The network batch size is the size of the random subset of reference data used to update the weights and network soft-decisions in a given iteration. This hyper-parameter is a natural way to control the communication overhead of our algorithm. As seen in Fig. 4, changing the network batch size affects the performance of D-Distillation. In deviation from the proofs of this work, separately adjusting the step size schedule for the private and distillation loss terms will be necessary to properly account for different batch and data set sizes on the private and reference data sets. Studying the full effect of the network batch size will enable the tuning of the accuracy and communication tradeoff to meet a desirable working point for the application at hand.

*Model size*: Since D-Distillation does not communicate a vector that scales with the weights of the DNN, model size can be scaled up to meet the devices' computational capabilities without affecting the per iteration communication overhead. Given that a larger DNN will impact the time to convergence for D-Distillation, the communication overhead reduction for larger DNNs requires detailed experiments.

*Model variety*: As opposed to D-SGD, D-Distillation can include model variety in its communication network of devices. Varying the architectures of DNNs has been shown to improve performance and mitigate overfitting in ensemble learning [17]. Our experiment in Fig. 3 confirms these results. More extensive simulations on larger networks with more classifier architectures, even beyond DNNs, are expected to exhibit larger performance gains over D-SGD.

*Compression*: D-Distillation significantly reduces the communication overhead before employing soft-decision compression. Applying compression methods was not the focus of this paper, which focused instead on the theoretical concept of D-Distillation. However, our experiment in Fig. 4 shows that off the shelf communicated reduction schemes can be easily applied to our communicates soft-decisions. Recently, more sophisticated gradient compression schemes such as sparsification combined with gradient correction have demonstrated a great deal of communication reduction (see for example [27, 28]). It is highly likely that a careful design and combination of these more sophisticated communication reduction schemes can easily reduce the communication overhead of D-Distillation further, with little to no performance degradation.