[Reviews · NeurIPS 2020]

Review 1

Summary and Contributions: The paper presents a decentralized learning approach that - similar to distillation - uses a regularization term in the local loss that forces local models to make similar predictions. For that, the predictions of all models are propagated through the network, similar to a distributed consensus algorithm. The paper provides a convergence guarantee for the proposed approach and empirically evaluates it in a limited simulated environment.

Strengths: - The approach is interesting, since it "synchronizes" local models not by aggregating their model parameters, but by forcing them to make similar predictions. This allows for more flexibility both in heterogeneity of local models and local minima. - The approach is applied to a decentralized setting where not all nodes are connected. - The theoretical analysis is sound.

Weaknesses: - The empirical evaluation is fairly limited. The approach is tested on up to 8 local devices/nodes on a single dataset (CIFAR-10). In these experiments it is not clear if the approach is beneficial (see additional feedback for details). - The convergence analysis only proves that eventually all local models converge to a critical point, but it provides no convergence rate.

Correctness: - in Sec. 3.1, the communication of the proposed approach and federated averaging is compared. There, it is stated that the communication of the proposed approach scales only with the number of classes K. As I understand it, the soft decision for every data point has to be communicated, though. Thus, it scales with K times the total dataset size. Since in distributed learning, the dataset size is typically much larger than the number of parameters of the model (not for MNIST, but in practice it usually is the case), in such cases the amount of communication of the proposed approach seems to be larger than that of federated averaging. -> I was wrong here. As the authors pointed out in their rebuttal, only a part of the decisions on the reference dataset is shared. I would ask the authors to make this more clear in a future version of the paper. - In Sec. 2, after Eq. (1), the authors claim that the Lipschitz-continuity of \nabla L_n would imply that the gradient is bounded. This is not the case: the function f(x) = x^2 has a Lipschitz-continuous gradient f'(x) = 2x, but the gradient is not bounded. I guess, the authors meant to demand that L_n is Lipschitz itself, in which case the gradient would indeed be bounded. -> I was mistaken about this point. As the authors pointed out in their rebuttal, s and y are bounded so that it can be shown that \nabla L_n has a bounded gradient.

Clarity: The paper is well-written, the approach is presented clearly, and the theoretical analysis is understandable and sound.

Relation to Prior Work: The proposed approach adds a regularization term to the local loss similar to distillation that forces local models to make similar predictions. However, in distillation the goal is to "distil" a large network into a smaller one, whereas in this paper this is not the goal. Thus, I would argue that terming it distillation is a bit misleading. In fact, the proposed approach is much closer to co-regularization [1,2,3]. [1] Brefeld, Ulf, and Tobias Scheffer. "Co-EM support vector learning." Proceedings of the twenty-first international conference on Machine learning. 2004. [2] Kumar, Abhishek, Avishek Saha, and Hal Daume. "Co-regularization based semi-supervised domain adaptation." Advances in neural information processing systems. 2010. [3] Brefeld, Ulf, et al. "Efficient co-regularised least squares regression." Proceedings of the 23rd international conference on Machine learning. 2006.

Reproducibility: Yes

Additional Feedback: - the empirical results do not look very convincing: the performance of distributed distillation is significantly worse than plain distributed SGD. The amount of communication required is substantially smaller, but comparable gains have been reached by federated averaging with C << 1 [3] or by dynamic averaging [4] with (seemingly) far better model performance (on a fully connected network graph, though). I suggest comparing to those baselines on a fully connected network. On a not-fully connected network I suggest comparing to decentralized learning approaches [5,6]. - calling the proposed approach a generalization of federated averaging is misleading, since the approach enforces similar outputs, not similar model parameter. The authors might argue that this has an advantages over federated averaging for non-convex problems: in federated averaging, averaging two models in different minima can lead to a resulting model that is way worse than each of the two local models. In the proposed approach, this could still be the case. However, if both models represent roughly the same function (e.g., if one is a reparameterization of the other), then the proposed approach will not change the models and thus does not deteriorate the performance. - why is the Euclidean distance between prediction vectors used as regularizer? Using a loss function (e.g., the KL-divergence) between the prediction vectors seems more flexible. - why is the approach limited to multi-class classification? It seems like it could straight-forwardly be applied to regression, as well. - while I do understand that computing power is a limiting resource, 8 nodes is far too little to analyze a distributed algorithm. If indeed the computing hardware does not allow for more nodes when training resnet, I suggest adding experiments with linear models. This allows to simulate far greater numbers of nodes. - it is great that the approach allows for different local models. It would be good to see the impact of different local model classes in the experiments. I suppose, there would be a trade-off, since all local models still need to be able to make the same predictions on all data. - The following is no criticism, but rather a comment: the use of the terms distributed, decentralized, federated, and parallel is not homogeneous in the community and there is a lot of ambiguity in all of them. Still, I would argue that the proposed algorithm is decentralized (as in, there is no central coordinator, e.g., like in federated learning), rather than distributed. I think the work is very promising and I encourage the authors to continue their work on it. However, I feel like the paper is not yet ready for publication. [3] McMahan, Brendan, et al. "Communication-efficient learning of deep networks from decentralized data." Artificial Intelligence and Statistics. 2017. [4] Kamp, Michael, et al. "Efficient decentralized deep learning by dynamic model averaging." Joint European Conference on Machine Learning and Knowledge Discovery in Databases. Springer, Cham, 2018. [5] Hegedűs, István, Gábor Danner, and Márk Jelasity. "Gossip learning as a decentralized alternative to federated learning." IFIP International Conference on Distributed Applications and Interoperable Systems. Springer, Cham, 2019. [6] Koloskova, A., Stich, S. & Jaggi, M.. (2019). Decentralized Stochastic Optimization and Gossip Algorithms with Compressed Communication. Proceedings of the 36th International Conference on Machine Learning, in PMLR 97:3478-3487 ---------------------- After Author Response ---------------------------------- I want to thank the authors for their detailed response, in particular for clarifying the amount of communication. This makes me more confident that the issues in the submitted manuscript can be fixed without major edits. Consequently, I have increased my score.


Review 2

Summary and Contributions: This work proposes a novel approach to federated learning that involves applying distillation in the distributed setting, without the need for data sharing. The idea is to have a reference dataset for a particular task, and have "workers" share their softmax class weights (on the reference dataset) with their peers, instead of sending the entire model or gradient updates. This helps reduce communication involved and allows "workers" training different DNN architectures to participate in the same training process, as long as they are working on the same task (with the same number of classes). The "workers" also keep their own data private by only sharing results on a reference dataset. The authors analyze an algorithm for this setup that works in two steps, 1) does a descent step using the worker's private data and softmax decision on the reference data, 2) shares new softmax decisions with neighboring workers. They show that their proposed algorithm reaches a particular stationary point, and also provide experiments demonstrating the utility of their algorithm.

Strengths: The idea of applying distillation to the federated learning space is certainly novel, and could lead to a lot of savings in communication in that setting.

Weaknesses: The algorithm proposed converges to a stationary point asymptotically. No rates are derived for the convergence. In simulations, there is a loss of accuracy compared to a full model sharing approach. It is unclear if this is just a limitation of the algorithm, or a greater limitation of distillation. More experiments with a larger network batch size may have helped to clarify this somewhat.

Correctness: Yes

Clarity: Yes, the paper is fairly well written and easy to understand

Relation to Prior Work: Yes, the related work is clearly discussed and the authors point out the differences of their work in relation to previously studied settings.

Reproducibility: Yes

Additional Feedback: There is an assumption of smoothness of the classification model. Is this common ? And are there any examples of real models that satisfy this ? #### After author response #### Thank you for taking the time to address my comments. I have read the response and other reviews and I think the newer experiments do help alleviate some of the concerns around comparison with D-SGD. I will keep my score unchanged.


Review 3

Summary and Contributions: To mitigate the issues of communication complexity and identical model architectures in decentralized learning, this paper developed a distributed distillation algorithm that enables the devices to learn from the soft-decision outputs given an unlabeled reference data shared commonly. In the paper, it was proved that the proposed algorithm converges with probability 1 to a stationary point for a smooth but non-convex function. The experimental results showed that compared with the baseline methods, the proposed method can outperform in terms of communication complexity, while underperforming in the test accuracy. This paper addressed an interesting topic in the decentralized learning via a different perspective, namely distillation, different from most existing methods where a global model was eventually achieved.

Strengths: The theoretical grounding in this paper is decent as it has sufficient contents for problem formulation, necessary definitions, and theoretical convergence analysis. The relevance to the NeurIPS community is also high, particularly in decentralized machine learning domain.

Weaknesses: The empirical evaluation in this work is unfortunately limited to support and validate the proposed algorithm. The significance and novelty of the contribution seems also limited as some claims in the paper look questionable. Please see the additional comments.

Correctness: The empirical methodology is correct though the evaluation is limited. While the claims and method are questionable.

Clarity: The paper is well written and easy to follow.

Relation to Prior Work: Overall yes. But I think the authors can have more literature survey on the decentralized learning area with some other existing algorithms.

Reproducibility: Yes

Additional Feedback: Though this paper is well written and easy to follow, there are several major issues that need to be addressed to improve the current form of the paper. 1. The experimental results were not convincing to validate the proposed algorithm. First, the accuracy gap between D-SGD and distributed distillation is large. According to Figure 2, we can see that ultimately the test accuracy for D-SGD is 85% while it is 80% for the distributed distillation 128. This difference is so significant that I doubt the distributed distillation could perform well when the dataset is more complex, such as CIFAR 100 or ImageNet. Furthermore, though we can see the communication complexity is reduced for distributed distillation 16 or 128 compared with D-SGD, it should be noted that this happened when the test accuracy was lower. No evidence has been shown in the paper that whether the distributed distillation could maintain the low communication complexity or not when reaching the same accuracy of D-SGD. The same conclusion applies to the results in the supplementary materials. Also, there are many other decentralized learning algorithms existing in the literature survey. As baseline methods, it would be better to see more comparison. 2. The use of unlabeled reference data. I think this could be difficult to determine, particularly the size. In the paper, the authors mentioned 7%, but for different datasets, how to determine the correct value? The authors have initially investigated the impact of the reference data on the performance, but with only one dataset, which is not sufficient. Also, according to Lemma 5, when the size of reference data is larger, the upper bound of the gradient for either the device or the soft-decision is larger. Why is the test accuracy in Figure 4 higher when the reference data size is larger? 3. The graph defined in the paper is strongly connected and directed. However, the way defined for W seems no difference as done in the connected and undirected graph. Would the theoretical results also apply to connected and undirected graphs? Additionally, the number of devices in the graph to validate the proposed algorithm is small (only 8) and no clear graph has been defined, cycle graph, or other kinds of graphs? 4. One of motivations for the paper is that distributed distillation can be model architecture-agnostic. It looks like in the experiments the author still defined the same ResNet-14 model for all devices. Then this issue is still remaining after the development of the proposed algorithm. I think the authors should run experiments with really different models in different devices. 5. According to the Definition 1, the authors required the algorithm to converge to (4) which is the first-order necessary condition for EACH devices. Then how can the authors guarantee that each stationary point for each device is local minimum, instead of local maximum or saddle point? For the distributed distillation, it seems more complicated in convergence to most existing consensus-based learning algorithms that aim to achieve a global model. In the distillation, each device needs to be taken care of. 6. Line 228 needs more clarification. Lines 233 and 234 claim that the average in (9) is dominant over the gradient step intuitively. I wonder how the authors obtained such an intuition. Not the reversible saying? If the model architectures were significantly different such that every \theta for each device was completely different, the average wouldn’t be easy to obtain and the gradient step could steer the dynamics away from the average. 7. Does the second moment in Lemma 5 need an expectation there? The auxiliary lemmas in the supplementary materials are trivial, in particular Lemma 8 and Lemma 9. They can be removed. In line 551, what are Section 12.2.2 and Section 12.2.3? After reading the feedback from the authors, I am more positive about the paper and then update my score. I would encourage the authors to continue working on it to make a solid paper.


Review 4

Summary and Contributions: The authors propose distributed distillation method for federated learning. Different from traditional federated learning that needs to communicate large amount of model parameters, distributed distillation only communicates soft-decisions of with common reference dataset that significantly reduces the communication. The contribution of this paper is as follows: a distributed decentralized distillation is proposed to reduce large amount of communications, each private device can train a different model architecture given the same classification space, proof of convergence is given in the paper.

Strengths: The idea is novel in general. Distillation is proposed as a model compression method, and this paper takes distillation as a information sharing method that extends the application of distillation. Distributed Distillation algorithm is proposed in this paper. In detail, each mobile device will distill knowledge from its neighbor's knowledge through reference data and learn the consensus of network soft-decisions. The communication overhead is low compared with traditional federated learning and the authors also provide a convergence analysis which is also a contribution of this paper.

Weaknesses: - The paper should do more experiments on more tasks to show the effectiveness of the proposed method, such as tasks in NLP, Speech. - The reference data may be hard to obtain for different applications although image classification task reference dataset is easy to get. In addition, practical way to deploy extra dataset on mobile devices should be considered since the memory limitation and hard disk storage are limited in real world application that I think should be discussed seriously.

Correctness: the method is empirically right. The empirical methodology is correct w.r.t to the authors claim.

Clarity: The paper can be written more clearly for model updates.

Relation to Prior Work: this work is different from previous contributions. Traditional federated learning works on communication of model parameters while the main point of this paper is to communicate the prediction of each devices, and the model structure can be different in this setting. In knowledge distillation area, the application in federated learning has not been discussed before.

Reproducibility: No

Additional Feedback: One question about the definition of neighborhood devices in this work. Do you pre-define the neighborhood structure before learning on devices? or the neighborhood can be adaptive during runtime.

[Author Response · NeurIPS 2020]

We thank the reviewers for their detailed and constructive comments, especially during these unprecedented times.

**Our Motivation (R1,R3,R4,R5)**: is to enable on-device learning on power and bandwidth limited devices, where
the communication reduction offered by current state-of-the-art compressed D-SGD is a good start, but is simply not
enough for large DNNs with millions of parameters. It is therefore necessary to develop a strictly **complementary**
source of communication reduction. Our paper provides a proof of concept for such a new source of communication
reduction, establishing the theory and validating it on DNN experiments. Our algorithm isn't designed to compete (or
be compared) with **other decentralized algorithms (R1, R4)** but instead to be combined with them to substantially
reduce the communication overhead further. Our algorithm can be combined with: communicating only every $\tau$ gradient
updates (**R1** [3]), adaptively communicating only updates of significant magnitude (**R1** [4]), and compression of the
communication (**R1** [5,6]). We have expanded our literature review to discuss these algorithms (**R4**) and our interplay
with them. Our new experiment in Fig. A below shows that by combining (simple) communication reduction schemes
with our method one can operate under significantly stricter bandwidth constraints than was previously possible. For
the revised paper, we will test additional communication reduction schemes on D-Dist.

**Comparison to D-SGD (R1,R3,R4)**:Since on-device is a constrained setting there might be no choice but to lose some
performance compared to the ideal D-SGD. However, in our new experiment in Fig. B we achieve close to D-SGD
performance on MNIST with **16 nodes (R1, R4)**. Our new experiment in Fig. C demonstrates the gain from **different**
**local models (R1, R4)** our method obtains, which can outperform D-SGD in practice by allowing more devices into
the training process. Even by just joining two subsets with different models, our method matches D-SGD which is
forced to run on each subset of 4 devices separately. We will add to the paper an experiment with 4 different models.

Figure: A) We apply the schemes of only communicating every $\tau$ gradient steps ($\tau = 10$) and compression via quantization (8 bit) and TopK (keep the top $K = 3$ elements in each soft decision) included in (R1 [3,6]) to D-Dist. In the revised paper, we will add the CIFAR10 counterparts for experiments A) and B), and both the epoch and communication plots for all experiments.

**Communication (R1)**: Our algorithm communicates soft-decisions on the reference dataset (network soft-decisions)
*but only* communicates a "network batch size" number of soft-decisions (even as low as 16) at each iteration. Analogous
to how mini-batch SGD operates, we can tune the "network batch size". Our experiments show that we *save 100-600$\times$*
*(**improved to 10,000$\times$ in Fig A**) in communicated bytes* versus D-SGD when *comparing at the same accuracy level*.

**The Reference Dataset (R4, R5):** should as large as possible, up to device constraints similar to how we would select
the training dataset size. While the gradient bounds in Lemma 5 increase, performance improves since the set of
distillation stationary points gets more selective as devices have to agree on more data points (see experiment in Section
12.2.2). Reference data can be synthetic and then it is easy to obtain (as in co-regularization, see R1's comment).

**Lipschitz Continuous Gradients (LCG)(R1):** This holds since $\mathbf{s}, \mathbf{y}$ are bounded (line 133). If $\mathcal{L}_n(\mathbf{s}, \mathbf{y})$ has LCG (for
every $\mathbf{y}$) then $\mathcal{L}_n(\mathbf{s}, \mathbf{y})$ has bounded gradients since $\|\nabla_s \mathcal{L}_n(\mathbf{s}, \mathbf{y}) - \nabla_s \mathcal{L}_n(\mathbf{s}_0, \mathbf{y})\| \leq \|\mathbf{s} - \mathbf{s}_0\| \leq M (\nabla \mathcal{L}_n(\mathbf{s}_0, \mathbf{y})$
is a constant). Then $\mathcal{L}_n(\theta^n)$ is LC as a sum of compositions of LCs: $\mathcal{L}_n(\mathbf{s}, \mathbf{y})$ and $\mathbf{s}(\theta, x)$. $\nabla \mathcal{L}_n(\theta^n)$ has LCG from
the chain rule, since $\mathbf{s}(\theta, x)$ has LCG. We now explain that in detail. Simply assuming that $\mathcal{L}_n(\theta^n)$ is LC is also fine.
**Graphs (R4, R5)**: Our results apply to undirected graphs, a special case of a directed graph where strong connectivity
coincides with connectivity. The graph is pre-defined. It can be generalized to a time-varying graph with a more
cumbersome analysis. The graphs in this work were randomly drawn for a given maximum number of degrees per node.
**Euclidean Distance & Smoothness (R1, R3):** Our analysis needs the loss function to be Lipschitz smooth in both
variables; KL-divergence is not. A smooth model is a common assumption in theoretical analyses ([4,7,9] in our paper).
**First-order Necessary Condition (R4)**: We prove that devices not only converge, each to a local stationary point, but
that they also agree on the reference soft-decisions. A device that accidentally converged to a local maximum or saddle
point will suffer from poor performance and wouldn't produce the correct soft-decisions to agree with all other devices.
**Lines 228, 233-234 (R4)**: If $\mathbf{z}_t^n$ is a probability vector $\forall n$ then so is $\tilde{W}\mathbf{z}_t$ (stochastic matrix) and the sum over
$\mathbf{z}_t^n(x) - \mathbf{s}(\theta_t^n, x)$ is zero, making $\mathbf{z}_{t+1}^n$ a probability vector for all $n$. The averaging step is a matrix multiplication that
if operating "in a vacuum", has a geometrical convergence rate (Lemma 7). The norm of the consensus error is $O(\eta_t)$
(Lemma 4). In comparison, even centralized SGD has an error of $O(\frac{1}{\sqrt{t}})$ at best (non-convex), slower than $O(\eta_t)$.
**The "second moment" bound of Lemma 5 (R4):** holds with probability 1, so it does not need an expectation.

We now refer to the highly relevant line of work of **Co-Regularization (R1)**. Our theory applies to **Regression (R1)**
with minor modifications, and that's a very nice insight that we now discuss in the paper. Thanks!
**Minor Comments:** We have fixed all minor issues. Section 12.2.2 starts at line 589 and Section 12.2.3 at line 617.

[Meta-Review · NeurIPS 2020]

The topic under investigation was deemed of sufficient interest and significance to be presented and NeurIPS, and the author response helped remove some doubts from the reviewers.